



# On the simulations of aerosol pH in China using WRF-Chem (v4.0): sensitivities of aerosol pH and its temporal variations in haze episodes

Xueyin Ruan[1], Chun Zhao[1,2,3,*], Rahul A. Zaveri[4], Pengzhen He[5], Xinming Wang[6], Jingyuan Shao[7], Lei Geng[1,2,3,*]

[1]School of Earth and Space Sciences, University of Science and Technology of China, Hefei 230026, Anhui, China
[2]CAS Center for Excellence in Comparative Planetology, University of Science and Technology of China, Hefei 230026, Anhui, China
[3]Frontiers Science Center for Planetary Exploration and Emerging Technologies, University of Science and Technology of China, Hefei, China
[4]Atmospheric Sciences and Global Change Division, Pacific Northwest National Laboratory, Richland, WA 99352, USA
[5]School of Environment and Tourism, West Anhui University, Lu'an 237012, Anhui, China
[6]Guangzhou Institute of Geochemistry, Chinese Academy of Sciences, Guangzhou 510640, Guangdong, China
[7]Flight branch, Civil Aviation University of China, Tianjin 300300, China

*Correspondence to*: Lei Geng (genglei@ustc.edu.cn) and/or Chun Zhao (chunzhao@ustc.edu.cn)

**Abstract.** Precise estimation of aerosol pH in chemical transport models (CTMs) is critical to aerosol modeling and thus influencing policy development. We reported WRF-Chem simulated $PM_{2.5}$ pH over China during a period with heavy haze episodes in Beijing, and explored the sensitivity of the modeled aerosol pH to factors including emissions of nonvolatile cations (NVCs) and $NH_3$, aerosol phase state assumption and heterogeneous production of sulfate. We found default WRF-Chem could predict spatial patterns of $PM_{2.5}$ pH over China similar to other CTMs, but with generally lower pH values largely due

to the underestimates of alkaline species (NVCs and $NH_3$) and the difference in thermodynamic treatments between different models. Increasing $NH_3$ emissions in the model would improve the modeled pH in comparison with offline thermodynamic model calculations of pH constrained by observations. In addition, we found that aerosol phase state assumption and heterogeneous sulfate production are important in aerosol pH predictions for regions with low relative humidity (RH) and high anthropogenic $SO_2$ emissions, respectively. These factors should be better constrained in model simulations of aerosol pH in

the future. Analysis of the modeled temporal trend of $PM_{2.5}$ pH in Beijing over a haze episode revealed a clear decrease in pH from $5.21 \pm 0.88$ in clean period to $3.56 \pm 0.49$ in heavily polluted period. The increased acidity in more polluted conditions is largely due to the formation and accumulation of secondary species including sulfuric acid and nitric acid, even though being modified by alkaline species (NVCs, $NH_3$). Our result suggests that $NO_2$ oxidation is unlikely to be important for heterogeneous sulfate production in Beijing haze as the effective pH for $NO_2$ oxidation of S(IV) is at higher pH of ~6.





# 1 Introduction

The acidity of atmospheric particles plays an essential role in various chemical and environmental processes. Acidified dust particles can largely enhance the solubility of transition metals which may act as nutrients in oceanic ecosystems (Meskhidze et al., 2003), affecting global biogeochemical nutrient cycles (Kanakidou et al., 2018). The dissolved metals can
also generate reactive oxygen species, causing aerosol toxicity and adverse health effects (Fang et al., 2017). Particle acidity can strongly affect gas-particle partitioning of volatile and semi-volatile species such as $NH_3$, $HNO_3$, $HCl$ (Keene et al., 2004; Guo et al., 2017a), as well as organic acids and bases (Ahrens et al., 2012). Moreover, particle acidity is linked to aerosol chemical reactivity by altering aqueous-phase reaction rates which are important for secondary aerosol formation. Both laboratory experiments (Gao et al., 2004; Surratt et al., 2007) and field studies (Rengarajan et al., 2011) have demonstrated
that higher acidity could facilitate  production of secondary organic aerosol (SOA) from oxidation of volatile organic compounds (VOCs) due to an acid-catalyzed mechanism. In addition, aerosol acidity also significantly affects reaction mechanisms and rates of heterogeneous sulfate production (Seinfeld et al., 2006). As one of the most abundant inorganic components in fine particles, sulfate is considered to be a key driver for the severe haze events in China (Cheng et al., 2016; Wang et al., 2016). Therefore, a thoughtful understanding of aerosol pH variability and its precise prediction are important to
understand and quantify the formation rates and mechanisms of sulfate in Chinese haze using models, providing insights on the outbreak of the haze events.

However, for nowadays, aerosol pH is poorly constrained due to difficulties in direct measurement techniques (Freedman et al., 2019; Keene et al., 1998). Instead, thermodynamic models, such as ISORROPIA II (Fountoukis and Nenes, 2007), Model for Simulating Aerosol Interactions and Chemistry (MOSAIC) (Zaveri et al., 2008), and Extended Aerosol Inorganics
Model (E-AIM) (Clegg et al., 2003) are commonly used to calculate aerosol pH (Pye et al., 2020). These models typically predict particle deliquescence, gas-particle mass transfer, solid-liquid phase equilibrium, activity coefficients and aerosol water content (AWC) (Zaveri et al., 2008; Jia et al., 2018) under observed or modeled meteorological conditions and atmospheric chemical compositions. Some of these thermodynamic models have also been implemented in 3D chemical transport models (CTMs) for representation of aerosol processes. For example, the ISORROPIA II model is incorporated in many 3D models,
such as the Goddard Earth Observing System with Chemistry model (GEOS-Chem), the Community Multiscale Air Quality Modeling System (CMAQ) and the PM-CAMx, while MOSAIC is employed in the Weather Research and Forecasting Model coupled with Chemistry (WRF-Chem).

CTMs are useful tools to understand relevant physicochemical atmospheric processes and to formulate air quality management strategies. The reliability of particle acidity prediction in CTMs is crucial for aerosol modeling, especially for
modeling of secondary aerosol formations, and therefore has implications for policy development. Vasilakos et al. (2018) demonstrated that pH bias simulated by CMAQ can induce nitrate partitioning bias and thus influences the response of $PM_{2.5}$ composition to emission changes in the model. Using GEOS-Chem model with prescribed particle pH values, Shao et al. (2019)





investigated the impact of particle pH on heterogeneous sulfate production and found that the model predicts different relative contributions of sulfate formation pathways to total atmospheric sulfate burden under different pH conditions. Furthermore, a

recent review paper (Pye et al., 2020) highlighted the critical role of particle pH in model simulations of a variety of atmospheric chemical species and/or processes, as aerosol pH directly influences the chemical composition of aerosols as well as the reactivities of aerosol components.

Given the importance of aerosol acidity in secondary aerosol formation and its implications for the outbreak of Beijing haze, many studies have assessed the acidity of aerosols in northern China using CTMs or offline thermodynamic models

constrained by observed gas and/or aerosol compositions (Cheng et al., 2016; Wang et al., 2016; Liu et al., 2017; Guo et al., 2017b; Song et al., 2018; Tan et al., 2018; Ding et al., 2019; Xie et al., 2020; Shao et al., 2019; Pye et al., 2020; Shi et al., 2019; Tao et al., 2020). Such models predicted a large range of aerosol pH (~3 to ~7) in northern China haze events with no general consensus. For example, Cheng et al. (2016) estimated high aerosol pH between 5.4 to 6.2 over the North China Plain (NCP) using ISORROPIA II in forward (i.e., gas plus aerosol phase measurements as inputs) and reverse mode (i.e., only

aerosol phase measurements as inputs), and Wang et al. (2016) estimated a near neutral aerosol pH of ~7 over Beijing using the same model with a stable state assumption. These two studies proposed that the high aerosol pH was driven by the neutralizing effect of high levels of ammonia over northern China, and as a result, $NO_2$ oxidation of dissolved S(IV) was suggested to be the dominant heterogeneous sulfate formation pathway. However, not only the conclusion on the role of $NO_2$ oxidation in sulfate production (e.g.,(He et al., 2018; Shao et al., 2019)), but also the predicted aerosol pH during the haze

events was challenged by later studies (Liu et al., 2017; Ding et al., 2019; Tan et al., 2018). In particular, Liu et al. (2017) and Guo et al. (2017b) argued that increasing $NH_3$ does not lead to ambient aerosol pH to near neutral and aerosols should be always acidic (pH = 4.2–4.5) over Beijing regardless the level of ammonia using ISORROPIA II with a metastable state assumption. Furthermore, Song et al. (2018) pointed out that the high pH values estimated by ISORROPIA II in previous studies were in fact caused by code errors when the stable state assumption was applied. Song et al. (2018) further calculated

aerosol pH for winter Beijing of ~4.6 and ~4 on average using ISORROPIA II and E-AIM in forward mode, respectively, similar to the results estimated by Liu et al. (2017) and Guo et al. (2017b). Tan et al. (2018) and Ding et al. (2019) also indicated similar acidic aerosols with average pH values between 3 and 4.5 in Beijing using ISORROPIA II. Moreover, Shi et al. (2019) reported an observationally constrained aerosol pH of 3.4 ±0.5 for Tianjin using ISORROPIA II.  Using the GEOS-Chem model, Shao et al. (2019) estimated the mean aerosol pH was 4.3 (ranged from 3.0 to 5.4) for autumn and winter Beijing.

Using the CAMQ model, Pye et al. (2020) predicted mean aerosol pH of 4.5 ±0.8 for February Beijing, and an annual mean pH of 3.1 ±1.5 for Tianjin, while Tao et al. (2020) found that the mean aerosol pH was 5.4 in NCP during January of 2013 by using WRF-Chem coupled with ISORROPIA II, which is higher than results from the aforementioned studies except that of Cheng et al. (2016) and Wang et al. (2016).

WRF-Chem configured with MOSAIC is one of the most extensively used regional air quality models, and has provided

insights on meteorological and physicochemical processes & mechanisms regarding air pollution issues in China (Huang et



al., 2014; Chen et al., 2016; Du et al., 2020; Sha et al., 2019). Pye et al. (2020) indicated that aerosol pH predicted by WRF-Chem with the MOSAIC thermodynamic scheme is in reasonable agreement with observationally constrained pH estimates over the contiguous United States. However, the performance of WRF-Chem configured with MOSAIC on aerosol pH prediction in China remains rarely reported and evaluated by far. In this study, we used WRF-Chem configured with MOSAIC

to investigate aerosol pH over China during a few haze episodes (15 October 2014 to 02 November 2014, i.e., in the preceding weeks of the Asia-Pacific Economic Cooperation summit period) when extensive observational data were available. We explored the sensitivity of the modeled aerosol pH to aerosol cation composition, aerosol phase assumption/configuration, heterogeneous sulfate productions, etc., compared the modeled results with that estimated using offline ISORROPIA II constrained by observed and modeled gas-aerosol compositions, and discussed the spatiotemporal variability of the predicted

aerosol pH over China during the study period. The results should provide insights into the predictability of aerosol pH using WRF-Chem and improve the understanding of aerosol pH variability in Beijing and other regions in China.

## 2 Methodology

### 2.1 Model configuration

#### 2.1.1 The WRF-Chem model

In this study, the version (v4.0) of WRF-Chem updated by the University of Science and Technology of China (USTC version of WRF-Chem) was used. Compared to the publicly released version of WRF-Chem, the USTC version includes some additional capabilities such as contribution analysis of aerosol related processes and improved turbulent mixing of aerosols (Zhao et al., 2013a; Zhao et al., 2013b; Du et al., 2020). The model configurations used in this study were summarized in Table 1. The Carbon Bond Mechanism version Z (CBMZ) (Zaveri and Peters, 1999) and the Model for Simulating Aerosol

Interactions and Chemistry (MOSAIC) (Zaveri et al., 2008) with eight bins were used as gas-phase and aerosol chemistry modules, respectively. The Noah land surface model (Chen and Dudhia, 2001) and the Yonsei University (YSU) planetary boundary scheme (Hong et al., 2006) were used to represent land surface processes and boundary layer turbulent mixing, respectively. The Rapid Radiative Transfer Model for General Circulation (RRTMG) (Iacono et al., 2008) was used to calculate the longwave and shortwave radiations.

**2.1.2 MOSAIC**

MOSAIC is an aerosol model with sectional approach to represent aerosol size distribution. It includes treatments for simulating aerosol physical and chemical processes such as nucleation, coagulation, gas-particle partitioning and heterogeneous chemistry. The chemical species treated by MOSAIC include sulfate, nitrate, chloride, methanesulfonate,





carbonate, ammonium, sodium, calcium, black carbon, organic mass, and liquid water. Potassium and magnesium are

represented by equivalent amounts of sodium, while other unidentified inorganic species are gathered as "other inorganic mass" (OIN). The gas-phase species comprising $H_2SO_4$, MSA, $HNO_3$, HCl, and $NH_3$ are capable of partitioning into the particulate phase. MOSAIC consists of three submodules pertinent to the calculation of size-resolved aerosol pH as described below.

The Multicomponent Taylor Expansion Method (MTEM) is used to estimate the mean activity coefficients of various inorganic electrolytes in multicomponent solutions based on its values in pure binary solutions of all the individual electrolytes

present in the solution (Zaveri et al., 2005b). Zdanovskii-Stokes-Robinson (ZSR) mixing rule (Zdanovskii, 1948; Stokes and Robinson, 1966) is applied for calculation of aerosol water content. Most of the MTEM and ZSR parameters are derived from the comprehensive Pitzer-Simonson-Clegg (Pitzer and Simonson, 1986; Clegg et al., 1992) model at 298.15 K for self-consistency.

The Multicomponent Equilibrium Solver for Aerosols (MESA) (Zaveri et al., 2005a) uses a pseudo-transient continuation

method to solve the solid-liquid phase equilibrium reactions expressed as pseudo-transient precipitation and dissolution reactions. The equilibrium solution is determined by integrating the resulting stiff nonlinear ordinary differential equations until the system reaches the steady state.

The gas-particle partitioning module ASTEM (Adaptive Step Time-split Euler Method) is coupled with the thermodynamic module MESA-MTEM to solve the mass transfer equations (Zaveri et al., 2008). To reduce the stiffness, it

first separates the non-volatile from semi-volatile gases in the numerical solver. For non-volatile gases ($H_2SO_4$ and MSA), ASTEM analytically integrates the condensation for all size bins, while for semi-volatile gases ($HNO_3$, HCl and $NH_3$), it numerically integrates condensation and evaporation for all size bins. Since the gas-particle mass transfer rates are strongly affected by the phase state of particles, different procedures are selected in ASTEM for completely solid, completely liquid, and mixed-phase particles.

In completely liquid or mixed-phase particles, the $H^+$ ion molality ($mH^+$) is needed for mass transfer calculations. In order to determine $mH^+$, two domains, i.e., sulfate rich and poor domains, are defined by sulfate ratio, $X_t$.

$$X_t = \frac{C_{NH_4^+} + C_{Na^+} + 2C_{Ca^{2+}}}{C_{SULF} + 0.5C_{CH_3SO_3^-}} \qquad (1)$$

Where C represents specie concentration in liquid phase, and $C_{SULF} = C_{SO_4^{2-}} + C_{HSO_4^-}$. In the sulfate-rich domain (i.e., $X_t < 2$), the liquid phase tends to absorb negligible $HNO_3$ and HCl due to the high acidity, thereby suppressing the oscillation behavior

of $H^+$ concentration during numerical integration. In this case, the equilibrium $mH^+$ is calculated by explicitly solving the partial dissociation of the bisulfate ion together with the electroneutrality equation (Zaveri et al., 2005b). In the sulfate-poor domain (i.e., $X_t \geq 2$), the use of equilibrium $mH^+$ will cause oscillations in the numerical solution associated with the condensation and/or evaporation of $HNO_3$, HCl and $NH_3$. Therefore, a new concept of dynamic $mH^+$ was introduced, which is a function of equilibrium constants, mass transfer coefficients, and the gas and particle-phase concentrations of all the related


species (Zaveri et al., 2008). In this approach, the surface equilibrium equations and acid-base coupled condensation approximation are solved simultaneously to determine the dynamic $mH^+$ in each size bin.

## 2.2 pH calculation

The pH is defined as the negative logarithm of the hydrogen ion activity in an aqueous solution, following the recommendation by the International Union of Pure and Applied Chemistry (IUPAC).

$$\text{pH} = -log_{10}a_{H^+} = -log_{10}\gamma_{H^+}H_{aq}^+ \tag{2}$$

where $a_{H^+}$ is the activity of hydrogen ion in aqueous solution on a molality basis, $\gamma_{H^+}$ is the hydrogen ion activity coefficient (in this study assumed to be unity) and $H_{aq}^+$ is the hydrogen ion molality in particle liquid water (mole kg$^{-1}$, moles of H$^+$ ions per kg of solvent). As MOSAIC outputs size-resolved hydrogen ion molality, the pH of PM$_{2.5}$ in the model was calculated

using the following equation:

$$pH_{pm2.5} = \frac{\sum_i mH_i^+ \times W_i}{\sum_i W_i} \tag{3}$$

where $mH_i^+$ (mole kg$^{-1}$) is the hydrogen ion molality in size bin $i$, and $W_i$ (kg m$^{-3}$) is the aerosol water content in that particular size bin. There are 6 size bins for PM$_{2.5}$.

## 2.3 Experimental design

In this study, simulations were performed at 36 km horizontal resolution with 139 (west-east) × 148 (south-north) grid cells covering the entire China as shown in Fig. S1. The simulation period was from 15 October 2014 to 02 November 2014 with the first 3 days used as model spin-up. This period was chosen because severe haze events occurred in Beijing and extensive observational data were available to constrain the model and evaluate the results. Initial and lateral boundary conditions for meteorological variables were derived from the European Centre for Medium-Range Weather Forecasts

(ECMWF) reanalysis data with a 0.703 °× ~0.702 ° horizontal resolution that are updated every 6 h (ERA-Interim dataset). The chemical initial and boundary conditions were provided by a quasi-global WRF-Chem simulation configured as described in Zhao et al. (2013a). Anthropogenic emissions were obtained from the Multi-resolution Emission Inventory for China (MEIC) at a 0.1 °× 0.1 ° horizontal resolution for the year 2015 (Li et al., 2017a; Li et al., 2017b). For emissions outside of China, the Hemispheric Transport of Air Pollution version-2 (HTAPv2) at 0.1 °× 0.1 ° resolution for the year 2010 was used (Janssens-

Maenhout et al., 2015).

All experiments conducted were listed in Table 2. In addition to a default WRF-Chem simulation (named as the ORIG scenario), we also conducted simulations to investigate the sensitivities of the modeled pH to variables including aerosol concentrations of nonvolatile cations (NVCs, such as Na$^+$, K$^+$, Ca$^{2+}$, Mg$^{2+}$), semi-volatile species (e.g., ammonia and chloride),





as well as aerosol phase state assumptions and heterogeneous sulfate production. These sensitivity experiments were named

as CTL1, CTL2, CTL3, CTL3meta, CTL3het_NoIs, and CTL3het_Is, respectively.

NVCs can strongly modulate aerosol acidity (Vasilakos et al., 2018; Kakavas et al., 2021). However, the default WRF-Chem significantly underestimated NVCs concentrations as compared with observations (Fig. S3a and Fig. S3b). The CTL1 experiment was thus conducted with modified cation speciation profiles constrained by observations. To better match the observed NVCs concentrations, we set the mass of $Ca^{2+}$ was 7.5% of dust and 10% of OIN, $Mg^{2+}$ was 0.8% of dust, and $Na^+$

and $K^+$ from OIN were 13% and 5%, respectively. Note that $K^+$ and $Mg^{2+}$ were converted to charge-equivalent $Na^+$ amounts since MOSAIC does not explicitly treat $K^+$ and $Mg^{2+}$.

Ammonia is one of the most important atmospheric alkaline species, and considered as a dominant factor causing higher aerosol pH in China than in the United States (Guo et al., 2017b; Ding et al., 2019). Previous studies indicated that $NH_3$ may be underestimated in current bottom-up emission inventories and using the MEIC inventory underestimated $NH_3$ emissions by

about 40% for the north China (Zhang et al., 2018; Wang et al., 2018; Kong et al., 2019). In experiment CTL2, the $NH_3$ emissions were multiplied by 2 and the others were the same as CTL1. On top of CTL2 simulation, we also conducted a chloride sensitivity simulation (i.e., CTL3) by increasing chloride emissions to improve the model prediction of aerosol chloride concentrations compared with observations.

Ambient aerosol phase state is uncertain and difficult to constrain experimentally or theoretically due to difficulties in

obtaining the efflorescence relative humidity (RH) for multicomponent salts. In general, aerosol can be treated as in metastable or stable state, where metastable means the aerosol solution is supersaturated and stable means crystallization of salts could occur once the solution reaches saturation. In MOSAIC, a flag called "hysteresis water content" ($W_{hyst}$) is transported to determine whether the particles at a grid point are on the stable or the metastable branch of the hysteresis curve. This is the default phase state determination method in WRE-Chem. To explore the effect of phase state determinations on the predicted

aerosol pH, on top of CTL3 we performed CTL3meta simulation in which the aerosol phase was fixed as metastable.

The last, aerosol pH can also be influenced by heterogeneous sulfate production as which is the main acid component of aerosol (Tilgner et al., 2021). We incorporated heterogeneous S(IV) oxidations in aerosol water into MOSAIC chemical mechanism using the same reaction parameterizations in Shao et al. (2019). The incorporated heterogeneous reactions include reactions of dissolved S(IV) with $H_2O_2$, $O_3$, $NO_2$ and $O_2$ catalyzed by transition metal ions (Table S1). Under this circumstance,

we also tested the effects of ionic strength on aerosol pH prediction as it influences heterogeneous sulfate production (Cheng et al., 2016; Liu et al., 2020). These two additional simulations on top of CTL3 were named as CTL3het_NoIs and CTL3het_Is, with the latter explicitly involved the effects of ionic strength on $H_2O_2$ and TMI-catalyzed S(IV) oxidations. In particular, for heterogeneous S(IV) oxidations, the first-order rate constant (k, $s^{-1}$) for the loss of gaseous species on aerosols was calculated by as follows (Jacob, 2000):

$$k = \left(\frac{R_p}{D_g} + \frac{4}{v\gamma}\right)^{-1} S_p \tag{4}$$





where $R_p$ is the radius of aerosol (cm), $D_g$ is the gas-phase molecular diffusion coefficient (cm$^2$ s$^{-1}$), $\nu$ is the mean molecular speed (cm s$^{-1}$), $\gamma$ is the uptake coefficient of SO$_2$ on aerosols (dimensionless), and $S_p$ is the aerosol surface area per unit volume of air (cm$^2$ cm$^{-3}$). The parameter $\gamma$ is obtained for each heterogeneous pathways using a similar method as Shao et al. (2019):

$$\gamma = \left[ \frac{1}{\alpha} + \frac{\nu}{4K^*RT\sqrt{D_aK_{chem}}} \cdot \frac{1}{f(q)} \right]^{-1} \tag{5}$$

where $\alpha$ is the mass accommodation coefficient (dimensionless), $K^*$ is the effective Henry's law constant (M atm$^{-1}$), R is the universal gas constant (L atm mol$^{-1}$ K$^{-1}$), T is air temperature (K), $D_a$ is the aqueous phase molecular diffusion coefficient (cm$^2$ s$^{-1}$), $K_{chem}$ is the first-order chemical loss rate constant in the liquid phase (s$^{-1}$), and $f(q)$ is given by:

$$f(q) = \coth q - \frac{1}{q} \tag{6}$$

$$q = R_p \left( \frac{k_{chem}}{D_a} \right)^{\frac{1}{2}} \tag{7}$$

**2.4 Observations**

The ground observations of inorganic components of PM$_{2.5}$ (SO$_4^{2-}$, NO$_3^-$, NH$_4^+$, Ca$^{2+}$, K$^+$, Na$^+$, Mg$^{2+}$, Cl$^-$) as well as the observed RH data were obtained from the HOPE-J$^3$A (Haze Observation Project Especially for Jing–Jin–Ji Area) field campaign located at the campus of the University of the Chinese Academy of Sciences (40.41 °N, 116.68 °E, around 20 m from the ground) which is around 60 km northeast of downtown Beijing (He et al., 2018; Yang et al., 2018; Chen et al., 2015;

Zhang et al., 2017). The aerosol composition data were used to evaluate the model's prediction on NVCs, and these data along with the observed RH were further used as inputs to calculate PM$_{2.5}$ pH using the ISORROPIA II model (in the forward and metastable mode). The ISORROPIA II model results were treated as observational constrained PM$_{2.5}$ pH and compared with that from the WRF-Chem simulations.

**3 Results**

**3.1 Spatial variability of simulated PM$_{2.5}$ pH**

Figure 1 shows the spatial distribution of the WRF-Chem predicted surface PM$_{2.5}$ pH over China averaged from 18 October 2014 to 02 November 2014 under default WRF-Chem configuration and a set of sensitivity experiments as listed in Table 2. The PM$_{2.5}$ pH was calculated by using weighted average aerosol water content as described in Sect. 2.2. The whole area of China was divided into six sub-regions (Fig. 1a) including the Taklimakan Desert (TD), the Gobi Desert (GD), the

Northeast Plain (NEP), the North China Plain (NCP), the Yangtze River plain (YR) and Southern China (SC) to review the spatial variability of the modeled pH.



In ORIG simulation (Fig. 1b), WRF-Chem predicted $PM_{2.5}$ pH with distinct spatial patterns, spanning ~0–7 pH units over China. The highest mean $PM_{2.5}$ pH is predicted over the GD (~4.18 ±2.23) and TD (~5.71 ±1.44), where nonvolatile cations (e.g., $Ca^{2+}$) from mineral dust is abundant, and the predicted pH is consistent with CMAQ and GEOS-Chem simulations of fine-mode aerosol pH (approximately 4–6) downwind of the deserts (Pye et al., 2020). Notably, the $PM_{2.5}$ pH shows a declined trend from the north towards the south, with mean pH values over NEP, NCP, YR and SC are 2.95 ±0.78, 2.29 ±0.39, 1.74 ± 0.38 and 1.66 ±0.29, respectively. Though the spatial features of $PM_{2.5}$ pH predicted by the default WRF-Chem model are similar with those from other chemical transport models (e.g., (Shao et al., 2019; Pye et al., 2020)), WRF-Chem generally tended to predict lower aerosol pH (0.8–3.6) over most regions of southern and Central China compared to other studies (1.3–5). For example, WRF-Chem predicted an averaged $PM_{2.5}$ pH of (2.3 ±1.3) for Beijing during the modeling period, which is 1–2 pH units lower than those reported by other studies using offline ISORROPIA II model constrained by observed aerosol and/or gas compositions (~3–4.5) for fall and winter Beijing (Tan et al., 2018; Song et al., 2018; He et al., 2018), and ~2 units lower than the GEOS-Chem predictions within the same period (Shao et al., 2019). The WRF-Chem model predicted $PM_{2.5}$ pH of ~2.2 in Tianjing is also lower than the values reported by Shi et al. (2019) who estimated the pH of $PM_{2.5}$ in Tianjing is ~3.4 using ISORROPIA II and ~3.1 using CMAQ. For a southern city, Guangzhou, WRF-Chem predicted the pH of $PM_{2.5}$ is ~1.2 ±1.0, lower than the estimate from Jia et al. (2018) (~2.5–2.8) but who reported values for July and used different models (ISORROPIA II, E-AIM IV and AIOMFAC).

To show the effects of the above-mentioned influencing factors on the predicted $PM_{2.5}$ pH, the differences in $PM_{2.5}$ pH between sensitivity runs were also displayed in Fig. 2. Compared to the ORIG run, the modeled $PM_{2.5}$ pH in the CTL1 run shows a ubiquitous increase all over China owing to the increased concentrations of NVCs in $PM_{2.5}$ (Fig. 2a). In particular, the $PM_{2.5}$ pH changes were more prominent over the NEP and NCP regions, where $PM_{2.5}$ pH increased by more than 0.9 pH units on average (Fig. S2). For regions near the deserts, i.e., GD and TD, $PM_{2.5}$ pH were increased by 0.8 and 0.7 pH units, respectively. In comparison, relatively small increases (~0.65 and ~0.47) in $PM_{2.5}$ pH were noted over YR and SC where aerosol was relatively acidic in the ORIG run (Fig. S2).

When $NH_3$ emissions were doubled (CTL2 scenario), the predicted $PM_{2.5}$ pH displayed diverse degrees of elevation (Fig. 2b), increased by 0.2–0.8 for most areas of China except for TD and GD where pH stayed nearly constant (Fig. 2b and Fig. S2). The rise in mean $PM_{2.5}$ pH was comparable (0.3–0.4) among NEP, NCP, YR and SC. In addition, minimal values of $PM_{2.5}$ pH showed slight increases (0.2–0.6) while the maximum values remained almost unchanged.

For the CTL3 scenario that included extra chloride emissions, the predicted $PM_{2.5}$ pH indicated negligible decreases compared to CTL2 (Fig. 2c), similar to the findings of Tao et al. (2020). Due to the low sensitivity of simulated aerosol pH to $Cl^-$ concentration, the result of CTL3 scenario and the potential effect of $Cl^-$ is not further discussed. However, it is noteworthy that WRF-Chem underestimated $Cl^-$ concentrations compared to the observations. In addition, $Cl^-$ is the precursor of reactive chloride species (e.g., Cl, $ClNO_2$, HOCl) that are important in atmospheric oxidation capacity (Wang et al., 2019; Wang et al.,





2020b). Therefore, future research should be devoted to the development of anthropogenic and natural chloride emissions to
improve the prediction.

With regard to CTL3meta scenario which specified the aerosol to be in metastable state indiscriminately, significant decreases (~1.2–1.8) in $PM_{2.5}$ pH compared to CTL3 were predicted over northwestern China and Tibet while the changes were smaller elsewhere (Fig. 2d). In particular, $PM_{2.5}$ pH decreased by ~1.87 for TD and ~1.13 for GD, causing pH down to 4.8 and 4, respectively, whereas the metastable state assumption had little impacts on the predicted $PM_{2.5}$ pH in the NCP, YR
and SC regions.

In the CTL3het_NoIs scenario, additional sulfate production (on top of CTL3) resulted in noticeable decrease of $PM_{2.5}$ pH over eastern and central China (Fig. 3a) where gas precursors (e.g., $SO_2$) from anthropogenic emissions are high. The largest decrease in the predicted mean $PM_{2.5}$ pH occurred in the NCP, by about 0.9 pH unit, compared with that of 0.7 pH unit in YR, 0.25 pH unit in SC and 0.17 pH unit in NEP (Fig. S2). However, $PM_{2.5}$ pH changes became negligible in TD and GD,
which may be attributed to their low $SO_2$ emissions and low abundance of AWC that limit local heterogeneous production of sulfate. $PM_{2.5}$ pH changes in CTL3het_Is scenario displayed spatial patterns similar to that of CTL3het_NoIs scenario but with smaller degree of decreases in $PM_{2.5}$ pH (Fig. 3b).

### 3.2 Temporal variation of $PM_{2.5}$ pH in haze events

During the study period, several haze episodes occurred over Beijing and there were several complete evolution cycles
of pollution level from very clean to severely polluted conditions. Over this period, time slots were referred to as "clean", "light pollution", "moderate pollution" and "heavy pollution" days according to different levels of $PM_{2.5}$ mass concentrations of 0–75, 75–115, 115–150, and >150 $\mu g\ m^{-3}$, respectively. To further investigate the evolution of $PM_{2.5}$ pH during a haze cycle, time series of the predicted $PM_{2.5}$ pH values over Beijing during the study period were shown in Fig. 4. The average values and ranges of $PM_{2.5}$ pH during the entire period, as well as the pollution levels were also listed in Table S2.
All the simulation results exhibit large but similar temporal variations in $PM_{2.5}$ pH during the study period, typically covering extreme acidic (<2) to alkaline (>7) pH levels (Fig. 4). As shown in Table S2, the largest pH range (0.64–7.63) was predicted by the CTL3het_NoIs scenario, and the smallest pH range fluctuating between 2.09 and 7.54 was found in the CTL2 scenario. The simulated pH from other scenarios varied by approximately 6 pH units. The large variations of $PM_{2.5}$ pH during haze episodes are consistent with the results from other studies. For example, He et al. (2018) utilized ISORROPIA II to
estimate $PM_{2.5}$ pH during Beijing winter haze and found a similarly large pH range of 3.4–7.6 when assuming metastable aerosol state. Gao et al. (2020) calculated aerosol pH in Tianjin using ISORROPIA II and reported that $PM_{2.5}$ pH ranged from −0.08 to 13.75, in which pH varied more severely.

On the other hand, similar temporal patterns of $PM_{2.5}$ pH were found in all scenarios, i.e., aerosols became more acidic at higher $PM_{2.5}$ levels (Fig. 4 and Table S2). During clean period, $PM_{2.5}$ pH spanned a wide range, with maximum pH values above 7 and minimum pH values below 2 (for ORIG, CTL1, CTL3het_NoIs) and below 2.5 (but above 2, for CTL2, CTL3,



CTL3meta, CTL3het_Is). For light pollution period, $PM_{2.5}$ pH exhibited a similar range as in the clean period, but with a lower mean value. However, under moderate and heavy pollution conditions, $PM_{2.5}$ pH was concentrated in a narrow range, varying within 1.5 pH units and with the most acidic aerosols (with mean pH values mostly between 1.5 and 3). These findings are consistent with those of Ding et al. (2019), who employed ISORROPIA II to calculate $PM_{2.5}$ pH in Beijing for four seasons

and found that the highest $PM_{2.5}$ pH appeared on clean days ranging from 2 to 7, followed by polluted and heavily polluted days for all seasons except winter. Analysis in Gao et al. (2020) also showed that the range of pH was more confined with aggravation of air pollution.

In Fig. 4, we also plotted the offline model results of $PM_{2.5}$ pH (termed as pH-obs) from ISORROPIA II (forward mode and metastable state) constrained by observed $PM_{2.5}$ compositions and RH. As gaseous $NH_3$ observations were not available,

so we estimated the values using an empirical equation following He et al. (2018). The observed $PM_{2.5}$ compositions were in coarse resolution (12 or 24 hours), so that the pH-obs results were also 12 or 24 hour averages. As shown in Fig. 4, pH-obs in general varied similarly to those predicted by WRF-Chem, but with higher absolute values. The default WRF-Chem (ORIG scenario) showed the maximum deviation (up to 2.22 pH units on average) from pH-obs. With the modifications of NVCs and $NH_3$ emissions, CTL2 scenario efficiently improved the discrepancies between WRF-Chem predictions and pH-obs (the mean

bias was reduced from 2.2 pH unit to 0.62). Similar discrepancies (~0.8 pH units) were found under CTL3meta and CTL3het_Is scenarios. The differences between other scenarios (i.e., CTL3 and CTL3het_NoIs) and pH-obs were larger than 1.2 pH units.

In addition, the responses of the predicted $PM_{2.5}$ pH to varying influencing factors under different pollution levels differed. The average pH of Beijing $PM_{2.5}$ was increased by 0.9 in CTL1 run compared to ORIG run, with the largest increase (~1.4)

found in clean period and smaller increases (~0.6) occurring in all other pollution periods. In contrast, the predicted pH increase (~0.36) was the smallest for the clean periods when $NH_3$ emissions were doubled (the CTL2 scenario), followed by light pollution (~0.92), moderate pollution (~1.11) and heavy pollution (~1.22) periods. Both increasing $Cl^-$ emission (CTL3 scenario) and changing phase state assumption (CTL3meta scenario) led to negligible impact on pH in Beijing among all periods. For the two additional scenarios that incorporated heterogeneous S(IV) reactions, when considering ionic strength

effects (CTL3het_Is scenario) little changes in the predicted $PM_{2.5}$ pH were seen, but more pronounced changes were seen when ionic strength effects were not taken into account (CTL3het_NoIs scenario). The latter case led to the decreases in pH by 0.7 and 1.3 units for moderate and heavy pollution periods, respectively, and increases by 0.3 units for clean period.

## 4 Discussion

Overall, the modeled $PM_{2.5}$ pH over China by all experiments displayed a clear spatial pattern, being more acidic in

Southern China while neutral in northwestern China. This spatial pattern is mainly controlled by dust emissions from the desert regions in northwestern China. In addition, the $PM_{2.5}$ pH appeared to be the most sensitive to the abundance of alkaline species

(i.e., NVCs and NH₃). In addition, for NCP where experienced severe and frequent haze events, PM$_{2.5}$ pH was also very sensitive to the magnitude of heterogeneous sulfate production; while for the TD and GD regions, the phase state assumption appeared to be important. In the discussions as follows, we analyzed the sensitivity of PM$_{2.5}$ pH to influencing factors, as well as the evolution of PM$_{2.5}$ pH in a haze development cycle in Beijing.

## 4.1 Sensitivity of the PM$_{2.5}$ pH spatial variability to influencing factors

### 4.1.1 The influence of NVCs

Aerosol composition (e.g., shifting in the relative fractions of anions versus cations) is known to influence its pH (Tao and Murphy, 2019; Lawal et al., 2018; Ding et al., 2019). NVCs are the alkaline components of aerosol which can neutralize sulfuric acid irreversibly and impact aerosol water amount through its effects on aerosol composition, thereby influence aerosol pH both directly and indirectly (Guo et al., 2018a; Vasilakos et al., 2018; Kakavas et al., 2021).

In the ORIG simulation, the model significantly underestimated the observed Ca$^{2+}$ and Na$^+$ concentrations in Beijing (Fig. S3a and S3b). Note Mg$^{2+}$ and K$^+$ are not included in the model but regarded as charge-equivalent Na$^+$, therefore the simulated Na$^+$ was compared to the observed sum of Na$^+$, K$^+$ and Mg$^{2+}$, while simulated Ca$^{2+}$ was directly compared with observed Ca$^{2+}$. As seen in Fig. S3a and S3b, Ca$^{2+}$ and Na$^+$ were significantly underestimated in the ORIG simulation by ~96.8% and ~97.6%, respectively, suggesting missing cation emission sources in model, which could lead to an underestimation in pH. To improve the model's performance in NVCs prediction, in the CTL1 run we modified the cation emission profile as described in Sect. 2.3. As a result, the simulated NVCs became more consistent with the observations, with a normalized mean bias (NMB) ≤ ± 5%.

Compared to the ORIG simulation, CTL1 predicted higher PM$_{2.5}$ pH almost everywhere with varying degrees as illustrated in Sect. 3.1. This is mainly due to the increased aerosol NVCs. In Fig. 5a, we plotted the changes in PM$_{2.5}$ pH in response to the changed aerosol NVCs as a function of the pH values from the ORIG simulation. The data were categorized in six sub-regions as indicate in Fig. 1a. As shown in Fig. 5a, the response of PM$_{2.5}$ pH to elevated NVCs displays a saddle-shaped curve. In all, for regions (e.g., NEP) with moderate acidic aerosol (i.e., pH = ~3–4) predicted by ORIG, their pH increased the most in response to elevated NVCs, indicating a large sensitivity of the aerosol pH to NVCs. While for regions with very acidic (e.g., in SC, pH ≤ ~1) or nearly neutral (e.g., in the central part of GD) aerosol pH, the response to elevated NVCs were minimum. This saddle-shaped curve response can be explained as follows. For aerosols with nearly neutral pH, they already contained high abundance of alkaline species (i.e., NVCs and/or ammonium), and addition of NVCs won't change their NVCs significantly. What is more, addition of NVCs may facilitate NH₃ partitioning to the gas-phase, lowering pH. As a result, little to no changes in pH should be expected. On the other hand, for very acidic aerosols with PM$_{2.5}$ pH < 2, the amounts of NVCs increase cannot reduce H$^+$ effectively due to excessive acids which may partition more to the aerosol phase to neutralize NVCs, and thus only exerted a small influence on aerosol pH. While for aerosols in intermediate pH ranges, there





were neither sufficient acidic species to neutralize the elevated alkaline NVCs, nor enough NVCs to buffer the added amount, so that the response was large. This effect was the largest for aerosols with pH around 3.

It is also noteworthy that, in this study the modified NVCs emission profiles were only constrained by observations in Beijing (located in the center of NCP) for the purpose of sensitivity test. This may be one of the reasons why the responses of $PM_{2.5}$ pH to elevated NVCs were the most in NCP and NEP which are closely located and influenced by the same dust emission sources. Nevertheless a more accurate NVCs emission inventory needs to be addressed in future model developments given the sensitivity of the modeled pH to the abundance of aerosol NVCs.

### 375    4.1.2 Sensitivity to NH₃ emissions

In addition to $Ca^{2+}$ and $Na^+$ (i.e., the NVCs) abundances, $NH_3$ is also an important alkaline component and plays essential role in aerosol pH by neutralizing acidic components ($H_2SO_4$ and $HNO_3$) to form particulate sulfate and nitrate and thus driving $NH_3$ towards to the particle phase (Wang et al., 2020a; Zheng et al., 2020; Zhang et al., 2021). But the $NH_3$ emission inventory used in WRF-Chem (MEIC) was suggested to be underestimated in China (Kong et al., 2019; Li et al., 2021). Therefore, CTL2
was performed to investigate the sensitivity of the modeled $PM_{2.5}$ pH to $NH_3$ emissions. After doubling $NH_3$ emissions, the response in $PM_{2.5}$ pH was not as large as that to NVCs. This is somewhat expected as in comparison with $NH_3$, NVCs can also neutralize acidic components but with a greater preference due to their low volatility. As a result, in regions close to the dust sources (i.e., in the northwest) or affected by dust outflows, the relatively high pH and sufficient NVCs tend to prevent the partitioning of $NH_3$ to aerosols, leading to limited response in $PM_{2.5}$ pH to $NH_3$ variation. As shown in Fig. 2b, in TD and GD,
$PM_{2.5}$ pH were increased negligibly and even somewhat decreased. While for regions with relatively low aerosol pH, more $NH_3$ can be partitioned to the aerosol phase to consume $H^+$, increasing pH. This is clearly seen in Fig. 5b where increases in $PM_{2.5}$ pH due to elevated $NH_3$ emissions were larger for more acidic aerosols. These results agree well with previous studies which have shown that pH responds nonlinearly to the changes in $NH_3$ emissions (Wang et al., 2020a; Ding et al., 2019; Liu et al., 2017) .

### 390    4.1.3 Sensitivity to aerosol phase state assumption

In chemical transport models, the history of the phase state of atmospheric aerosols cannot be easily tracked as aerosols move and mix quickly between different grid points due to turbulent transport (Zaveri et al., 2008). For this reason, it is challenging for models to determine whether the mixed aerosols follow the efflorescence branch (i.e., metastable state) or the deliquesced branch (i.e., stable state). When aerosols with different hydration histories and phase states mix together, the
resulting particles in a given size bin must all be placed either on the stable or the metastable branch of the hysteresis curve as the aerosol size distribution at a grid point is represented by a single set of size bins. In MOSAIC, the phase state of particles in different size bins can be different as the model determines whether the particles in a given size bin are on the stable or the





metastable branch using the $W_{hyst}$ parameter (Zaveri et al., 2008). In comparison, many previous studies investigated aerosol pH during Beijing haze events by assuming the aerosols are in metastable states, which is regarded as a reasonable assumption

for high RH (> 50%) conditions (Liu et al., 2017; Guo et al., 2017b; Guo et al., 2018b; Ding et al., 2019). ISORROPIA II adopted in some CTMs (e.g., GEOS-Chem, CMAQ) also applies the metastable state assumption (Shao et al., 2019).

As shown in Fig. 2d, after fixing aerosol phase to metastable, the response (decrease) of the modeled $PM_{2.5}$ pH was larger for regions with aerosols that were less acidic, especially for GD, TD, and central Tibet. These regions were also in general with low RH (Fig. S4). RH is known to affect AWC and thus the phase state of aerosols. Karydis et al. (2021) reported similar

findings in their modeling study, that the metastable assumption caused a pH decrease (~2 pH unit on average) over the regions with low RH and high crustal species. To explore the effects of the phase states on the predicted $PM_{2.5}$ pH, we plotted the pH of aerosols in each size bin (bin 01–bin 06 with increasing particle diameters from 0.039 to 2.5 μm) from CTL3 and CTL3meta runs in Fig. S5. The first impression from reviewing Fig. S5 is that the modeled decreases in $PM_{2.5}$ pH in CTL3meta were mainly caused by changes in the first four size bins. Notably, in the CTL3 run, aerosols in these bins (01–04) in GD, TD and

central Tibet were determined to be mostly solid (i.e., no liquid water thus no pH exists) due to low RH. But in the CTL3meta run when metastable state was assumed, these aerosols were calculated to have a very small amount of water (Fig. S6) and thus the pH were very low. Further analyses on the components of aerosols in these size bins (Table S3) in TD indicated that they were high in sulfate but low in NVCs, suggesting "sulfate rich" particles that are in general highly acidic (Zaveri et al., 2008).

For regions with RH > 70%, little to no changes in $PM_{2.5}$ pH were predicted when fixing aerosol phase to metastable. This is because that when RH > 70%, aerosols in all size bin may be already determined to be in metastable state by $W_{hyst}$ in the default MOSAIC scheme. In addition, since both states predict a liquid aerosol at ambient RH > 70% which reaches the deliquescence RH for most mixed-salt aerosols, changes in pH between stable and metastable states at higher RH should be insignificant as modeled. Our modeled results are also consistent with that from previous box model and chemical transport

modeling studies which found a similarly small effect of phase assumption on pH at high RH condition (Song et al., 2018; Tao et al., 2020). In all, these results demonstrate that metastable assumption might be inappropriate at low RH conditions and would lead to unrealistic pH predictions. This in turn suggests the rationality and advances of MOSAIC scheme in phase state determination in WRF-Chem.

### 4.1.4 Sensitivity to heterogeneous sulfate production

Sulfate is the main acidic component of aerosols and thus largely determines aerosol pH (Weber et al., 2016; Tilgner et al., 2021). We implemented the heterogeneous sulfate formation pathways on aqueous aerosols in WRF-Chem in this study, and explored the effects of ionic strength on the production rates with two additional runs, i.e., CTL3het_Is and CTL3het_NoIs. Overall, after the addition of heterogeneous S(IV) oxidations, modeled sulfate concentrations increased largely over eastern and central China (Fig. 3c), and where $PM_{2.5}$ pH decreased significantly as a consequence (Fig. 3a). This is as expected because





sulfate can release free $H^+$. On the other hand, the effects of sulfate production on pH can be buffered by uptake of bases (e.g., ammonia) from the gas-phase (Zheng et al., 2020), which could differ by regions depending on $NH_3$ level. AWC also changes in response to changes in aerosol components, which in turn affects aerosol pH. Therefore, $PM_{2.5}$ pH change in response to additional sulfate production in the system was in fact a result of the combination of these factors.

Notably, for the CTL3het_Is run, $PM_{2.5}$ pH changes were much smaller (Fig. 3b) compared to the CTL3het_NoIs run
because of a smaller amount of additional sulfate production (Fig. 3d). As reported by Liu et al. (2020), high ionic strength can largely inhibit the TMI-catalyzed reaction rate and slow it down by a factor of ~85 at an ionic strength of 2.8 M. Although high ionic strength would make the reaction of S(IV) with $H_2O_2$ faster in aerosol water (Liu et al., 2020), the modeled low $H_2O_2$ concentration hindered the contribution of this reaction to sulfate production despite the effects of high ionic strength. Therefore, when ionic strength was considered, the heterogeneous production of sulfate was inhibited and thus smaller
decreases in pH were caused. Note the inclusion of heterogeneous sulfate production here was just used to test the sensitivity of $PM_{2.5}$ pH to variations in acidic components, but not aiming to simulate atmospheric sulfate so that we did not conduct further analyses on the model's ability to capture observed sulfate production. Recent experimental studies suggest that interfacial chemistry at aerosol surfaces may also be important for ambient sulfate formation, such as the newly proposed aerosol-phase acceleration for the Mn-catalyzed oxidation of S(IV) (Wang et al., 2021) and water-assisted interfacial reaction
of $NO_2$ with $SO_3^{2-}$ (Liu and Abbatt, 2021). Inclusion of these additional sulfate formation pathways would presumably increase sulfate production and lower the modeled $PM_{2.5}$ pH further. However, large uncertainties still remain in atmospheric sulfate formation mechanisms especially for these newly proposed mechanisms, and the kinetic parameters in concentrated solutions (i.e., the surface of aerosols) also need to be accurately constrained by further investigations.

### 4.2 Driving factors of the temporal $PM_{2.5}$ pH variation in Beijing haze

As all modeled scenarios displayed a similar temporal variation for the studied period in Beijing, here we chose the CTL3meta scenario for further discussion on the temporal evolution of $PM_{2.5}$ pH and driving factors under different pollution levels. Figure 4 shows that the predicted $PM_{2.5}$ pH values were in general lower (more acidic) at more polluted days for all WRF-Chem simulations as well as the ISORROPIA II results constrained by observed aerosol composition and RH. To reveal this trend more clearly, the corresponding pH values in Beijing under different pollution levels modeled by the CTL3meta
scenario were illustrated in the box-and-whisker plots in Fig. 6a. In addition to the WRF-Chem predictions (Fig. 6a), the offline ISORROPIA II estimates using WRF-Chem outputs (i.e., aerosol composition and RH from CTL3meta scenario, Fig. 6b) and observations (Fig. 6c) were also displayed. Figure 6 illustrates that $PM_{2.5}$ pH calculated by ISORROPIA II (both based on WRF-Chem simulated data or observational data) generally shows consistent patterns as WRF-Chem simulation, and the $PM_{2.5}$ pH was higher during relatively clean days while the lowest during heavy pollution days. The multiple model average of $PM_{2.5}$
pH in Beijing under heavy pollution (> 150 μg m$^{-3}$) was 3.56 ± 0.49. These results suggest that $PM_{2.5}$ pH in Beijing under heavy haze conditions is likely moderate acidic, and thus the $NO_2$ oxidation pathway unlikely dominates in heterogeneous





sulfate production. As $NO_2$ oxidation of dissolved S(IV) only becomes effective in less acidic pH ranges (~6) (Cheng et al., 2016). Most recently, an experimental study (Liu and Abbatt, 2021) proposed a water-assisted interfacial mechanism for $SO_2$ oxidation by $NO_2$ at the aerosol surface that can maintain its atmospheric importance at a lower pH of 5. This value is

nevertheless still higher than the predicted pH during the heavy haze period and thus implying an unlikely importance of $NO_2$ oxidation.

In addition, we noticed that the high pH values were generally associated with high mass fractions of NVCs and low AWC, whereas low pH values were often accompanied by low mass fractions of NVCs and high AWC (Fig. S7). This suggests the important roles of AWC and aerosol compositions in determining $PM_{2.5}$ pH. To explore their relationship, mass fractions

of $PM_{2.5}$ ionic species as well as AWC under different pollution levels are shown in Fig. S8. As the pollution deteriorated, AWC increased and the mean value reached 88.0 μg m$^{-3}$ during the heavy pollution period (Fig. S8b). What is more, NVCs had a higher proportion of 0.19 in clean period, compared to 0.06 in light pollution period, 0.04 in moderate pollution period and 0.03 in heavy pollution period (Fig. S8a). This is consistent with changes in $PM_{2.5}$ pH as NVCs tend to increase pH. These results are in line with some previous studies (Ding et al., 2019; Shi et al., 2017) who have demonstrated that the role of NVCs

in aerosol acidity. But some other studies found NVCs have limited impacts on aerosol pH, which may be due to the relatively minor contribution of crustal ions to aerosol mass in their cases (Liu et al., 2017; Zheng et al., 2020; Zhang et al., 2021). In addition, the mass fraction of sulfate declined from clean periods (0.16) to light and moderate pollution periods (0.08) then slightly increased in heavy pollution periods (0.10). Nitrate had the predominant mass fraction, accounting for 0.49 during clean period and remaining almost constant during other periods (0.65). Sulfate and nitrate formation were apparently enhanced

on more polluted conditions. This lead to the release of free $H^+$ which promotes the partitioning of ammonia into the aerosol phase, neutralizing the formed acidic species and buffering the pH. This also at least in part explains why the mass fraction of ammonium increased steadily throughout the haze evolution with 0.10, 0.18, 0.20 and 0.21 for clean, light, moderate and heavy pollution periods, respectively.

Ambient RH has also been recognized as a key factor in the evolution of winter haze events (Tie et al., 2017; Sun et al.,

2013) and aerosol acidity (Tao and Murphy, 2019; Battaglia et al., 2017; Ding et al., 2019; Jia et al., 2020). This can also be seen in Fig. 4 where RH was in general high at more polluted days. Here we analyzed the correlation of AWC and pH with RH. As shown in Fig. 7, AWC exponentially increases with increasing RH, with a mean value of (0.018 ± 0.006) μg m$^{-3}$ at 20% RH and (130 ± 43) μg m$^{-3}$ at 100% RH. In contrast, $PM_{2.5}$ pH shows a general decreasing trend with RH. These can be explained as follows. RH is typically low at the start-up phase of haze events, under which condition NVCs from primary

aerosols would be rich and gas uptake as well as secondary aerosol formation are restricted due to the limited AWC, thereby leading to higher pH (clean period). As RH elevates with the deterioration of $PM_{2.5}$ pollution, greater amounts of AWC are formed caused by the acceleration of aerosol hygroscopic growth. AWC then serves as an efficient medium for heterogeneous reactions on the surface of aerosols, thereby substantially enhancing secondary formation of acid species (such as sulfate and nitrate) and resulting in greater acidity. The latter is also facilitated by the accumulation of reactive gas precursors as the haze





event evolves under stable boundary layer conditions. Aerosol hygroscopic growth is further enhanced by a positive feedback
mechanism that the production of secondary aerosol species can in turn enhance aerosol hygroscopicity and increasing AWC
(Wu et al., 2018). It should be noted that more AWC could also exert a dilution effect which would dilute the $H^+$, but the acid
effect likely prevails over the dilution effect leading to a net drop of pH.

### 4.3 Comparison of PM$_{2.5}$ pH predictions between MOSAIC and ISORROPIA II

To further explore the potential effects of different thermodynamic models on the modeled aerosol pH differences between
this study and previous studies, we also compared the MOSAIC results with those obtained from ISORROPIA II. The WRF-
Chem simulated hourly chemical concentrations along with temperature and RH in Beijing from CTL3meta scenario were
used as inputs to ISORROPIA II (forward mode, assuming metastable to be consistent with CTL3meta). Time series of aerosol
pH (bin 01–bin 06) predicted by the two different models are given in Fig. S9. Overall, ISORROPIA II and MOSAIC predicted
a similar temporal pH trend, but ISORROPIA II in general predicted higher absolute pH values than that of MOSAIC for all
particles with the size less than 2.5 μm. What is more, a regression slope of 0.87 between the calculated PM$_{2.5}$ pH by MOSAIC
and ISORROPIA II was found (Fig. S10). These findings are comparable to the results reported by Pye et al. (2020) who found
that with the same model inputs, a regression slope of 0.89 between the calculated pH from the box-model version of MOSAIC
and ISORROPIA II was obtained. Comparisons of the pH values predicted by MOSAIC and ISORROPIA in Zaveri et al.
(2008) also showed a similar phenomenon that ISORROPIA tended to predict higher values under same conditions. The
discrepancy between these two models may be attributed to the higher amounts of aerosol water content predicted by
ISORROPIA II relative to MOSAIC, as indicated in Fig. S11, despite both models use the same phase state assumption and
RH. Difference in other fundamental thermodynamic treatments, including activity coefficients, gas-particle partitioning
scheme and solution approach may also account for the final pH difference. Nevertheless, the exact causes of the differences
in pH predicted by these two models remain to be explored.

### 5 Conclusion

In this study, the performance of WRF-Chem configured with MOSAIC in predicted PM$_{2.5}$ pH over China was evaluated.
In particular, using the model, we assessed the evolution of PM$_{2.5}$ pH over a few haze episodes in Beijing from 18 October
2014 to 02 November. The results indicate default WRF-Chem could predict similar spatial gradient of PM$_{2.5}$ pH across China
compared to other CTMs as reported by previous studies. However, WRF-Chem in general yielded low pH (0.8–3.6) over
most regions compared to other models (1.3–5). This is mainly due to the model underestimations of NVCs concentrations,
with additional contributions from low model NH$_3$ emissions as well as inherent differences in thermodynamic representations.
The latter was further assessed by comparing against the corresponding pH predictions from offline ISORROPIA II using





WRF-Chem modeled aerosol composition and RH as inputs. Compared to ISORROPIA II values, pH calculated by MOSAIC

is consistently lower by 0.6 units on average, despite the pH variation trend matched quite well.

Further, six experiments were conducted to investigate the response in modeled $PM_{2.5}$ pH to varying NVCs, $NH_3$, phase state assumption and sulfate production over China. The model results show that pH sensitivity have substantial spatial heterogeneity. Elevated NVCs emissions caused ubiquitous increases in $PM_{2.5}$ pH with higher effects in NEP and NCP regions where original pH is in the moderate acidic range. For regions with high or low original pH, the effects from NVCs are minor.

Doubling $NH_3$ emission also led to an increase in $PM_{2.5}$ pH over most areas of China except for TD and GD where are characterized by high aerosol pH and sufficient NVCs. The effects of phase state assumption on pH were found to be minor at high RH conditions but large decrease in $PM_{2.5}$ pH can be induced at low RH conditions due to an unrealistic metastable phase state assumption. Additional formed sulfate in aerosol water tended to effectively decrease $PM_{2.5}$ pH over eastern and central China in a complex manner, due to the buffering effect of semi-volatile ammonia and the accompanied AWC change.

In addition, $PM_{2.5}$ pH evolution during haze cycles in Beijing was investigated. The results indicate that aerosols became more acidic as haze pollution accumulated, from $5.21 \pm 0.88$ in clean period to $3.56 \pm 0.49$ in heavily polluted period, due to both changes in aerosol components and meteorological conditions. Large mass fraction of NVCs was found to be responsible for the high aerosol pH during clean periods. The elevated AWC with increasing RH during polluted periods accelerated secondary aerosol formation (e.g., sulfate and nitrate), enhanced water uptake and further lowered pH.

In all, our study suggests that NVCs and $NH_3$ influence the predicted $PM_{2.5}$ pH the most at least in the WRF-Chem model, but currently the model cannot predict the abundance and variations of these species especially for $Ca^{2+}$ and $Na^+$. Future research efforts need to be undertaken to better constrain NVCs and $NH_3$ emissions in model to improve aerosol pH predictions. A priori assumption that aerosols are at either stable or metastable state in model simulations (e.g., in GEOS-Chem) maybe less accurate compared to WRF-Chem which applies a more rigorous and computationally expensive phase state determination

approach, especially for aerosol pH prediction for regions with low RH. Follow-up studies to including more accurate and up-to-date heterogeneous sulfate formation pathways in model would also be necessary. The last, more high resolution observation datasets at different locations may be necessary to further evaluate MOSAIC predictions of aerosol pH.

**Code and data availability**

The release version of WRF-Chem can be downloaded from http://www2.mmm.ucar.edu/wrf/users/download/get_source.html.

The modified version of WRF-Chem used in this study is archived on Zenodo at https://doi.org/10.5281/zenodo.6359417. The ERA-Interim reanalysis data from the European Centre for Medium-Range Weather Forecasts (ECMWF) for initial and boundary conditions can be downloaded from https://rda.ucar.edu/datasets/ds627.0/.





**Author contributions**

XR, LG and CZ designed the experiments, conducted and analyzed the simulations. XR, CZ, RZ, PH, XW, JS, and LG
contributed to the discussion and final version of the paper.

**Competing interests**

The authors declare that they have no conflict of interest.

**Acknowledgements**

This work is financial supported from the National Natural Science Foundation of China (Awards: 41822605, 41871051 and
41727901), the Fundamental Research Funds for Central Universities and the Strategic Priority Research Program of Chinese
Academy of Sciences (XDB 41000000). Pengzhen He acknowledges support from Natural Science Foundation of Anhui
Province (2008085QD184) and from West Anhui University (WGKQ202001007). Rahul A. Zaveri acknowledges support
from the Office of Science of the U.S. Department of Energy (DOE) as part of the Atmospheric System Research program at
Pacific Northwest National Laboratory (PNNL). PNNL is operated for DOE by Battelle Memorial Institute under contract DE-
AC06-76RLO 1830. The numerical calculations in this paper have been done on the supercomputing system in the
Supercomputing Center of University of Science and Technology of China. X.Y. R is grateful to Qiuyan Du and Mingyue Xu
from USTC for help in the use of the WRF-Chem model.



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





**Table 1.** Summary of model configurations.

| Description | Selection |
| --- | --- |
| Horizontal grid spacing | 36 km |
| Grid dimensions | 149 × 138 |
| Aerosol scheme | MOSAIC 8 bin |
| Gas-phase chemistry | CBM-Z |
| Long wave Radiation | RRTMG |
| Short wave Radiation | RRTMG |
| Cloud Microphysics | Morrison 2-moment |
| Cumulus Cloud | Grell-Devenyi |
| Planetary boundary layer | YSU |
| Land surface | Noah land-surface model |


**Table 2.** Numerical experiments conducted in this study.

| Name | Cation | NH$_3$ emission | Cl emission | Phase state | Sulfate production |
| --- | --- | --- | --- | --- | --- |
| ORIG | default | default | default | default[a] | default |
| CTL1 | modify | default | default | default | default |
| CTL2 | modify | ×2 | default | default | default |
| CTL3 | modify | ×2 | modify | default | default |
| CTL3meta | modify | ×2 | modify | metastable | default |
| CTL3het_NoIs | modify | ×2 | modify | default | Add het (No Is effect) |
| CTL3het_Is | modify | ×2 | modify | default | Add het (consider Is effect) |

[a]By default, in MOSAIC a flag called "hysteresis water content" (W$_{hyst}$) is transported to determine whether the particles are on the stable or the metastable branch.


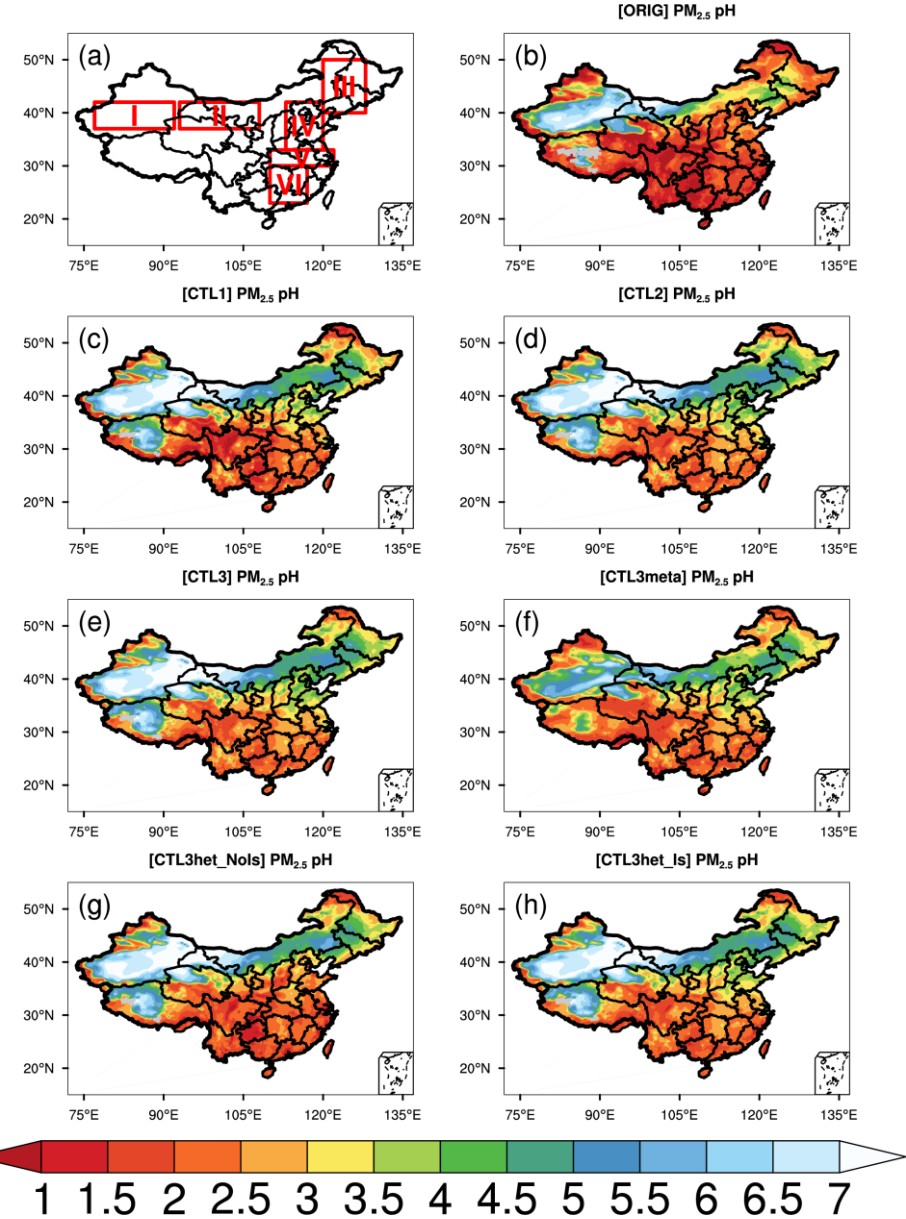

**Figure 1.** (a) Six sub-regions. (b-h) Spatial distributions of mean surface $PM_{2.5}$ pH (LWC-weighted average pH) during the study period of 15 October 2014 - 02 November 2014 predicted by (b) ORIG (c) CTL1 (d) CTL2 (e) CTL3 (f) CTL3meta (g) CTL3het_NoIs (h) CTL3het_Is. "I" in (a) represents the Taklimakan Desert (TD), "II" represents the Gobi Desert (GD), "III" represents the Northeast Plain (NEP), "IV" represents the North China Plain (NCP), "V" represents the middle and lower reaches of Yangtze River plain (YR), and "VI" represents Southern China (SC).


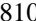


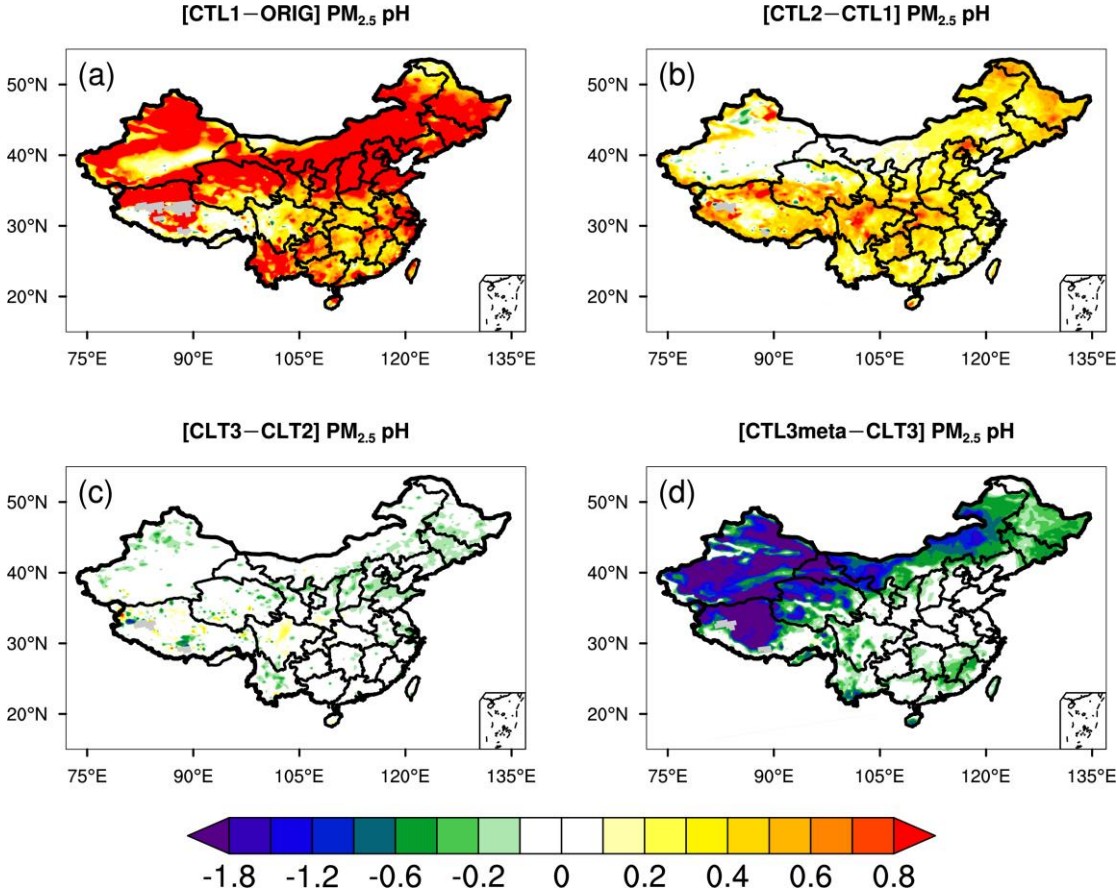

**Figure 2.** Spatial distributions of the difference in mean surface PM$_{2.5}$ pH during the study period of 15 October 2014 - 02 November 2014 between (a) CLT1 and ORIG scenarios, (b) CTL2 and CTL1 scenarios, (c) CTL3 and CTL2 scenarios, (d) CTL3meta and CTL3 scenarios.





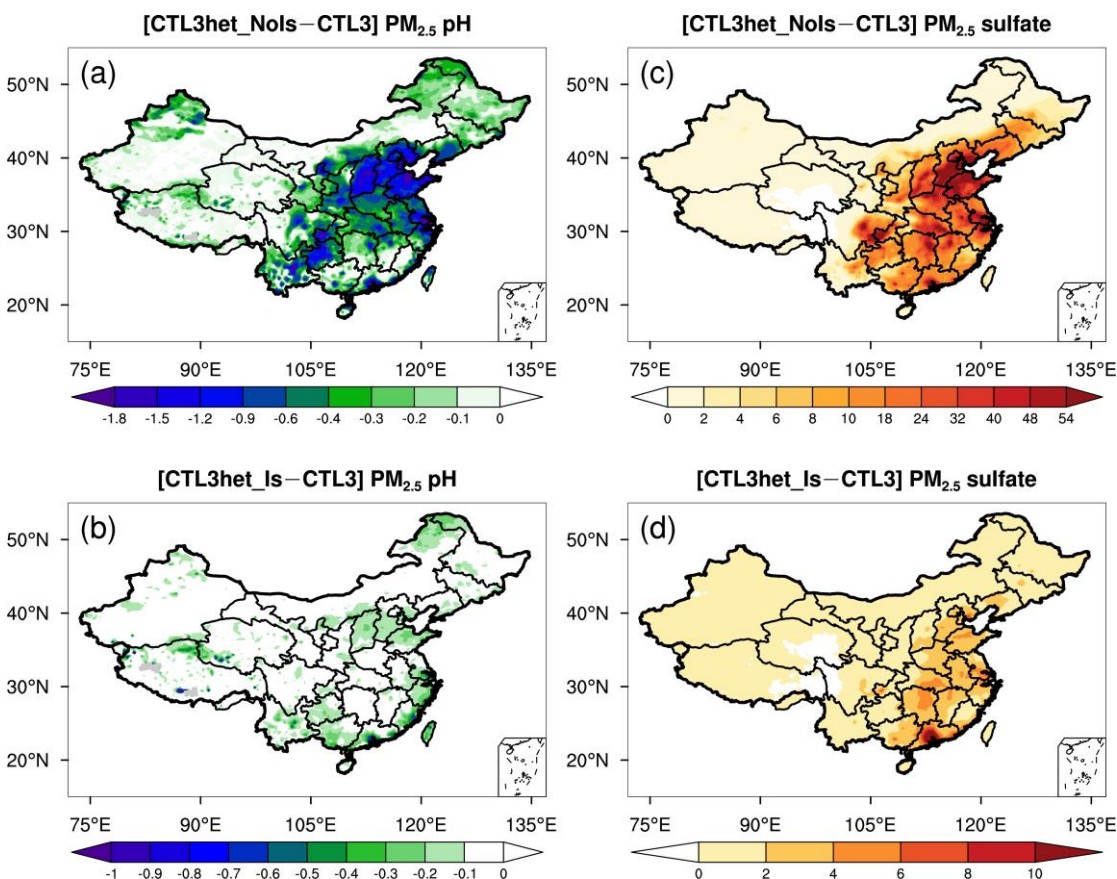

**Figure 3.** Spatial distributions of the difference in mean surface (a,b) PM$_{2.5}$ pH and (c,d) PM$_{2.5}$ sulfate (μg m$^{-3}$) between CTL3het_NoIs and CTL3 scenarios (top panels), and CTL3het_Is and CTL3 scenarios (bottom panels) during the study period of 15 October 2014 - 02 November 2014. Different scales are used.

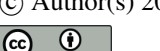



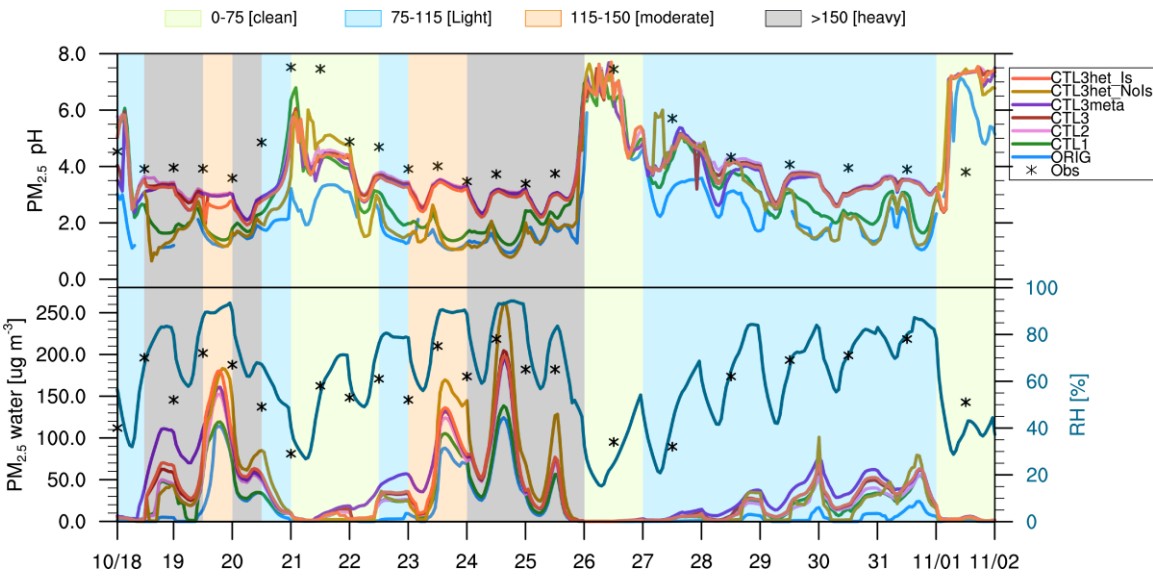

**Figure 4.** Time series of (top panel) surface $PM_{2.5}$ pH, and (bottom panel) $PM_{2.5}$ water contents (μg m$^{-3}$) (left y-axis) predicted by all WRF-Chem scenarios at Beijing site during the study period of 15 October 2014 - 02 November 2014 and Relative humidity (%) (RH, right y-axis) are given from ORIG scenario. ISORROPIA II-calculated pH values constrained by observations as well as the observed RH are shown as black star markers, with each value corresponding to a PM2.5 sample (12h or 24h). Shaded areas represent four different pollution levels (green-clean; bule-light; orange-moderate; grey-heavy).



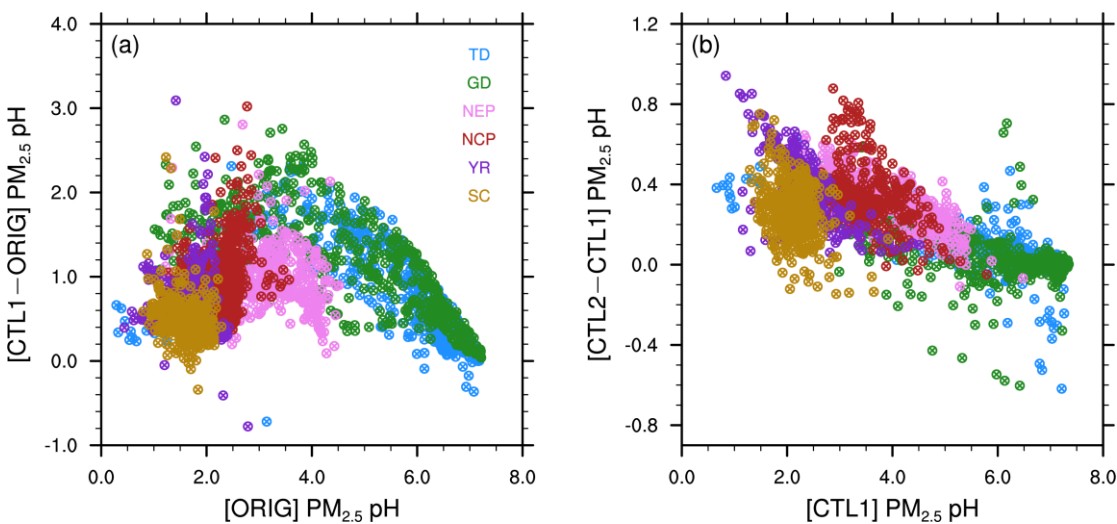

**Figure 5.** Scatterplots of the surface PM$_{2.5}$ pH differences between (a) CLT1 and ORIG scenarios, (b) CTL2 and CTL1 scenarios vs. the corresponding original pH, separated by regions.





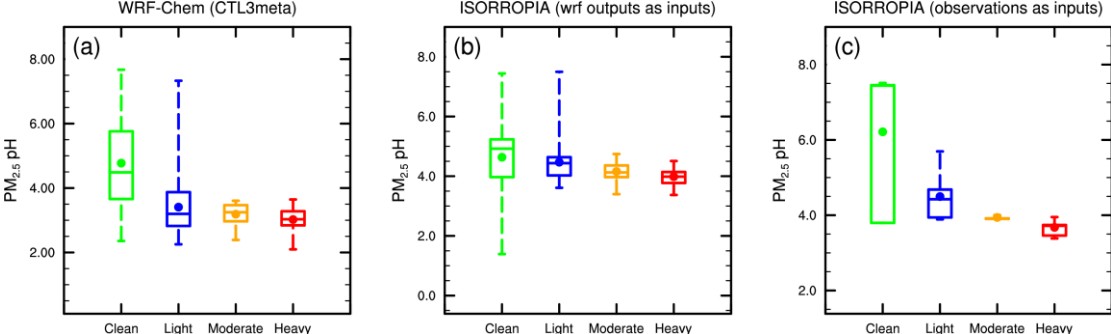

**Figure 6.** The box-and-whisker plots of surface $PM_{2.5}$ pH in each haze stage in Beijing from (a) WRF-Chem CTL3meta scenario, (b) ISORROPIA predictions with WRF-Chem (CTL3meta) relevant outputs as inputs, and (c) ISORROPIA predictions with observations as inputs. The boxes represent, from top to bottom, the 75th, 50th, and 25th percentiles of statistical data. The whiskers represent, from top to bottom, the minimum and the maximum, and the solid circles represent the mean values.





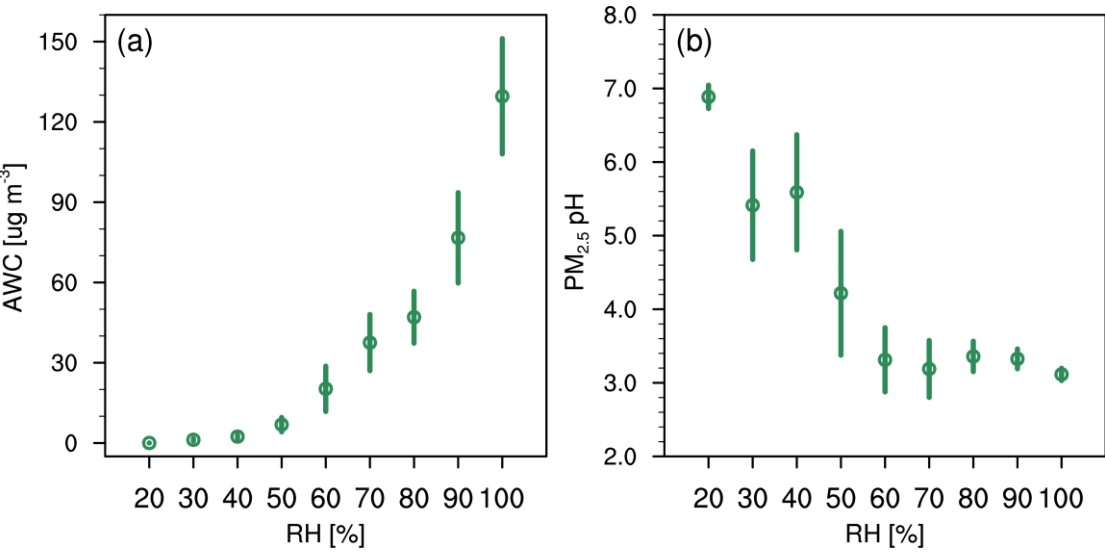

**Figure 7.** AWC (µg m$^{-3}$) (a) and PM$_{2.5}$ pH (b) predicted by CTL3meta scenario as a function of RH for data at Beijing site during the study period of 15 October 2014 - 02 November 2014. Data are grouped in RH bins (10% increment). The error bars represent the standard deviations.