# Peer review of "On the simulations of aerosol pH in China using WRF-Chem (v4.0): sensitivities of aerosol pH and its temporal variations in haze episodes"

_Geoscientific Model Development, 2022_

## Author Comment (AC1)

**Referee #1**

We are grateful to Dr. Song for his time reviewing this manuscript, and for the valuable comments. Below we present our detailed responses to Dr. Song's comments with the comments in italic and our response in normal font.

- *Geoscientific Model Development Discussions: "On the simulations of aerosol pH in China using WRF-Chem (v4.0): sensitivities of aerosol pH and its temporal variations in haze episodes". Ruan et al. investigates the aerosol pH simulations in WRF-Chem model focusing on haze episodes in Beijing. Overall, the paper is scientifically sound, clearly written and easy-to-follow. I only have a few minor comments:*

- *Line204: WRF-Chem*

**Response:** Thank you for pointing out this typo. It has been corrected.

- *Section4.1.1: I think the minimal response to elevated NVCs in the nearly neutral cases may be related to the role of carbonate. At least in E-AIM and ISORROPIA, the aerosol pH cannot get very high (8 or 10) because of the buffer role of carbonate.*

**Response:** We thank the reviewer for pointing out this. In the revised manuscript the role of carbonate was added in the revised text as "Further, carbonate could play a buffer role in keeping aerosol pH values from getting too high."

- *Section4.3: It is interesting to see a systematic difference between ISORROPIA and MOSAIC even when the model inputs are the same. The authors list several possible contributing factors including AWC, phase-partitioning method, activity coefficients and solution method. I suggest the authors to investigate this issue further: (1) is there any difference in the predicted gas-phase ammonia concentrations between the two models? This can give some clues on the contribution of phase partitioning method; (2) the authors note the large difference in AWC in the paper. It seems that both ISORROPIA and MOSAIC use the ZSR method. ISORROPIA uses a look-up table for hygroscopic curves of different electrolytes. One possible way to further examine the role of AWC calculation method is to replace the hygroscopic curves in ISORROPIA by the ones from WRF-Chem.*

**Response:** We think these are very good suggestions so we compared the predicted $NH_3$ concentrations between the two models and the results are shown in Figure R1 as attached. As we can see, there are negligible difference in predicted $NH_3$, suggesting that phase-partitioning method may not contribute to the pH difference. Therefore, in the revised manuscript we deleted the statements on phase-partitioning.

Regarding the second point, we agree with the reviewer that it is interesting to investigate the exact factors contributing to the systematic pH difference calculated by the two models. However, it requires substantial efforts to review the details of ISORROPIA code and seems out of scope of this study. In addition, ISORROPIA and MOSAIC differs in many ways and a comprehensive comparison could be the focus of future work. Note in the revised manuscript this section is moved to the supporting material as Text S1 follow the suggestion of reviewer#2 since more comprehensive comparisons between the two models' pH outputs has been done by Pye et al. (2020).

[Figure]

**Figure R1.** Time series of NH$_3$ concentrations predicted by WRF-Chem (CTL3meta, green line) and ISORROPIA II (black line) at the surface in Beijing. ISORROPIA II ("forward" mode, assuming metastable) was run with WRF-Chem simulated hourly chemical concentrations along with T and RH.

● *Figure5&6: it is good to show the different subplots in the same vertical scale or write a note to remind the different scales.*

**Response:** Thanks for the suggestion. For Fig. 5, in view of the large differences in y-axis values in different subplots, we kept the different scales and wrote a note as "Different scales are used." For Fig. 6 (now Fig. 8), we have made the vertical scales the same in each subplots.

**Reference**

Pye, H. O. T., Nenes, A., Alexander, B., Ault, A. P., Barth, M. C., Clegg, S. L., Collett, J. L., Fahey, K. M., Hennigan, C. J., Herrmann, H., Kanakidou, M., Kelly, J. T., Ku, I. T., McNeill, V. F., Riemer, N., Schaefer, T., Shi, G. L., Tilgner, A., Walker, J. T., Wang, T., Weber, R., Xing, J., Zaveri, R. A., and Zuend, A.: The acidity of atmospheric particles and clouds, Atmos. Chem. Phys., 20, 4809-4888, https://doi.org/10.5194/acp-20-4809-2020, 2020.

---

## Author Comment (AC2)

*Summary*
- *This paper discusses aerosol pH in China, examining the sensitivity of aerosol pH to non-volatile cations, ammonia (NH3) emissions, chloride (Cl-) emissions, aerosol aqueous phase chemistry, and assumed properties of the aerosol. Quantifying aerosol pH from observations and diagnosing its value in chemical transport models are needed to better understand impacts of aerosol pH on biogeochemical nutrient cycles and generation of reactive oxygen species that can cause aerosol toxicity and adverse health effects. Thus, the paper is an appropriate topic for GMD to publish.*

  *This is a very good paper and should make a substantial impact on understanding processes affecting aerosol pH. The paper is fairly well written, although several spots need to be clarified. My suggestions of needed clarifications are given below.*

**Response:** We are grateful to the reviewer for his time and the detailed and constructive comments. They are very helpful to improve the manuscript. Following the suggestions/comments of the reviewer, in the revised manuscript, we have added a few new figures in the supporting material to support some statements in the text and to address the comments. In particular, the spatial distributions of NVCs, $NH_3$, $Cl^-$ and $SO_2$ emissions were added. More information on how dust is treated and the default speciation profile were added in the methodology section. In addition, two sensitivity experiments at finer resolution (12 km) were performed to show the impact of grid spacing on aerosol pH. A schematic describing the temporal evolution of $PM_{2.5}$ pH during haze cycle was added. Other text and figures have also been revised as the reviewer suggested. Please find our point-by-point response (black) and the corresponding revisions (blue) below.

**Major Comments**
- *The authors conclude that non-volatile cations (NVC), ammonia (NH3) emissions, and chloride (Cl-) emissions influence the aerosol pH over different areas of China. However, there is no presentation of the spatial distribution of NVCs, NH3 emissions, or Cl- emissions. It would be useful to see the overlap regions of NVCs and NH3 emissions as well as the main locations for Cl- emissions and SO2 emissions. Please add these maps.*

**Response:** We thank the reviewer for the very good suggestion. Now we add Fig. S4 in the supporting material to show the spatial distributions of NVCs, $NH_3$, $Cl^-$ emissions from default configuration and the corresponding sensitivity experiments averaged over the study period. And we also add Fig. S6 to show the spatial distributions of $SO_2$ emissions. Now these figures are cited at appropriate positions in the main text to support our statements.

**Emissions**

[Figure]

**Figure S4.** Spatial distributions of emissions of (top panel) NVCs, (middle panel) NH₃, (bottom panel) and Cl⁻ from default configuration and its corresponding sensitivity experiment during the study period of 15 October 2014 - 02 November 2014.

[Figure]

**Figure S6.** Spatial distribution of SO₂ emission from the MEIC China inventory during the study period of 15 October 2014 - 02 November 2014.

● *Section 4 tends to repeat what was said in Section 3. Please review Section 4. I suggest writing it in a manner where the reader is reminded of the Section 3 result but spends more text focused on new information and discussion.*

**Response:** Thanks for this comment. In the original manuscript, we repeat a little bit the main results just to remind the readers of the results and then we start to discuss. Now we realize that this part is a bit reduplicated, and we have revised the text to reduce duplicate contents. For example, the second paragraph of Section 4.1.1 is removed, and the sentence "However, the NH₃ emission inventory used in WRF-Chem (MEIC) was suggested to be underestimated in China (Kong et al., 2019; Li et al., 2021). Therefore, CTL2 is performed to investigate the sensitivity of the modeled $PM_{2.5}$ pH to NH₃ emissions." in Section 4.1.2 is removed.

● *Do the authors think the aerosol pH results would differ much with finer horizontal resolution? I could imagine that the grid size of emissions and mesoscale flow patterns influences the range of aerosol pH. One of my comments below brings up this question again in terms of thinking of the average aerosol pH versus the range of aerosol pH during haze events and whether that range would change if the grid spacing were smaller.*

**Response:** We thank the reviewer for raising this. In response to this comment, we conducted two sensitivity experiments at finer resolution (12km) under ORIG and CTL3meta scenarios, respectively. Comparisons of modeled surface $PM_{2.5}$ pH at Beijing site during the study period are shown in Fig. R2. Overall, the patterns look very similar for both scenarios. This is likely due to the fact that the original grid box (36 km) is already very small, and within this resolution the spatial variabilities of emissions, air flow and etc. are probably already very small. Therefore, the results do not change the conclusion of this study that $PM_{2.5}$ pH in Beijing under heavy haze conditions is likely moderate acidic, and thus the NO₂ oxidation pathway highly unlikely dominates in heterogeneous sulfate production.

[Figure]

**Figure R2.** Comparison of surface $PM_{2.5}$ pH between 36km and 12km resolution runs based on (top panel) ORIN and (bottom panel) CTL3meta scenario at Beijing site during the study period of 15 October 2014 - 02 November 2014. Shaded areas represent four different pollution levels (green-clean; blue-light; orange-moderate; grey-heavy).

*Specific Comments*

● *I highly recommend using present tense instead of past tense.*

**Response:** Thank you very much for the point about the tenses; we have carefully checked the tenses in the whole manuscript and changed them to present tense.

● *Line 15, in abstract: Why is it critical to estimate aerosol pH accurately in chemical transport models as well as policy development? Please add a phrase.*

**Response:** Thanks for the comment. Now we revise the text and add a sentence. It now reads as: "Aerosol pH is a fundamental property of aerosols in terms of atmospheric chemistry and its impact on air quality, climate and health. Precise estimation of aerosol pH in chemical transport models (CTMs) is critical for aerosol modeling, and thus influencing policy development that partially relies on results from model simulations."

● *Section 2.1.1. Does the WRF-Chem simulation include observational nudging so that the meteorology is represented well?*

**Response:** In this study, the WRF-Chem simulations do not include observational nudging but are nudged to the reanalysis data, which was widely used to constrain regional meteorological simulation towards reanalysis. Now we add description of the nudging method in Section 2.3 as "The modeled u and v component wind, air temperature, and water vapor mixing ratio at layers above the planetary boundary layer (PBL) are nudged towards the reanalysis data with a 6 h timescale (Stauffer and Seaman, 1990; Seaman et al., 1995)." In addition, we add Fig. S2 in the supporting material to evaluate the simulated meteorological fields (winds at 850 hPa and 2m temperature). The results show that model can reproduce these basic meteorological fields with the spatial correlation coefficient of 0.99 and 0.98. We have clarified this in the revised manuscript as "The modeled winds at 850 hPa and temperature at 2m are compared with the ERA5 reanalysis dataset (Fig. S2), which show that the model can reproduce these basic meteorological fields with the spatial correlation coefficient of 0.98 and 0.99, respectively."

[Figure]

**Figure S2.** Spatial distributions of (a,b) wind speed at 850 hPa and (c,d) temperature at 2m from (left panels) ERA5 reanalysis datasets and (right panels) ORIG scenario averaged for the study period of 15 October 2014 - 02 November 2014.

- *Section 2.1.2. It would be good to add more information on how dust is treated in the model especially in relation to the ions associated with dust? Do iron, magnesium, and/or manganese affect the aerosol pH?*

**Response:** Thank you for this suggestion. The dust emission scheme we used is the adjusted GOCART dust emission scheme which has been fully coupled with the MOSAIC aerosol scheme in WRF-Chem (Zhao et al., 2010). Now we add descriptions of the dust emission scheme in Section 2.3 as "The Goddard Chemistry Aerosol Radiation and Transport (GOCART) dust emission scheme (Ginoux et al., 2001) is used to simulate natural dust emission fluxes, and the emitted dust particles are distributed into MOSAIC aerosol size bins based on the physics of scale-invariant fragmentation of brittle materials derived by (Kok, 2011). More details about the dust emission scheme coupled with MOSAIC aerosol scheme in WRF-Chem can be found in Zhao et al. (2010; 2013a)."

In the default dust emission scheme, only Ca and $CO_3$ emissions are scaled to dust emission flux with mass fraction of 1.2% and 1.8%, respectively. MOSAIC does not treat iron, magnesium, and manganese, thus these species have no effect on the modeled aerosol pH. However, as we state in the text, magnesium is treated as charge-equivalent amounts of sodium in this study.

- *Line 189. I am a little confused between dust and OIN in MOSAIC. I thought in WRF-Chem MOSAIC that dust is part of OIN (other inorganics). Therefore, it is not clear how the speciation profile is implemented when calcium, magnesium, sodium, and potassium are changed. It would also be good to know the speciation profile for the default simulation – please add a sentence in the second paragraph of Section 2.3. Please add a map to show where NVCs are high in concentration over China.*

**Response:** Sorry for the confusion. Dust is indeed part of OIN in publicly released version of WRF-Chem. However, dust and OIN are separated as a new feature in the USTC version of WRF-Chem as implemented by Zhao et al. (2010).

We have clarified this in Line 191 in the revised manuscript as "It is worth noting that dust and OIN are treated as two separate aerosol species in the USTC version of WRF-Chem.".

In the default simulation, the only source of Na is sea salt emissions, and Ca is scaled to dust emissions with a mass fraction of 1.2%. We have clarified this when presenting the reason for the NVCs underestimation in default simulation as "$Ca^{2+}$ and $Na^+$ are significantly underestimated in the ORIG simulation by ~96.8% and ~97.6%, respectively, because in ORIG simulation, the only source of $Ca^{2+}$ is scaled to dust emissions with a mass fraction of 1.2% and $Na^+$ is only from seasalt emissions.".

Spatial distributions of NVCs emissions from ORIN and CTL1 simulations are now displayed in Fig. S4a-b.

- *Lines 192-198, please add maps of NH3 and Cl- emissions. It would be informative to explain the emissions sources of ammonia and chloride. How much did Cl- emissions increase?*

**Response:** We thank the reviewer for the thoughtful suggestions. Now we add Fig. S4c-f in the supporting material to show the spatial distributions of $NH_3$ and $Cl^-$ emissions. In ORIG simulation, $Cl^-$ is only from seasalt aerosols; in CTL3 simulation, we include additional $Cl^-$ emissions with the mass fraction from OIN source of 15%. It is now clarified in Line 212-216 as "Figure S3c indicates that the modeled $Cl^-$ concentration is almost zero in ORIG simulation because there is only seasalt source of chloride and anthropogenic chloride emissions are not included. On top of CTL2 simulation, we conduct a chloride sensitivity simulation (i.e., CTL3) with additional emissions for chloride (assuming a 15% mass contribution from OIN) to improve the model prediction of aerosol chloride concentrations compared with observations."

- *Section 2.4, It is my understanding to calculate aerosol pH with ISORROPIA constrained by observations that gas-phase NH3 and HNO3 must be known. Could the authors please describe the measurements of NH3 and HNO3, or what values are used for these semi-volatile gases?*

**Response:** Yes, both gas and aerosol concentrations are needed when using forward mode of ISORROPIA. However, the observational data for $NH_3$ and $HNO_3$ are unavailable in the present study, so that $NO_3$ input is aerosol $NO_3^-$ only and gaseous $NH_3$ concentrations are estimated by a empirical equation following He et al. (2018). Now these points are clarified in Section 2.4 in the revised manuscript as "As gaseous $NH_3$ and $HNO_3$ observations are not available, we use aerosol $NO_3^-$ only as $NO_3$ input and estimated gaseous $NH_3$ values using the empirical equation $[NH_3]$ (nmol $mol^{-1}$) = $0.34\times[NO_x]$ (nmol $mol^{-1}$) + 0.63 following He et al. (2018)." And we further acknowledge this in the conclusion section as "In addition to aerosol composition, concurrent measurements of gas species subject to phase partitioning (e.g. $HNO_3$ and $NH_3$) will provide better constraints on acidity estimates."

Furthermore, we test the effects of uncertainties in $NH_3$ concentration on aerosol pH predictions by considering ±10% fluctuations in $NH_3$ concentration. The results show that ±10% changes in $NH_3$ concentration only result in +0.03 and -0.04 pH unit changes, respectively. Now we add these discussions as "In order to assess the effects of uncertainties in $NH_3$ concentration on aerosol pH predictions, we also run ISORROPIA II with ±10% fluctuations in $NH_3$ concentration and find little changes (i.e., +0.03 and -0.04 pH unit) can be induced. "

In addition, temperature is also needed to calculate aerosol pH which we missed in the original manuscript. This has been corrected in the revised manuscript.

- *Lines 243 and throughout the manuscript. It seems that the number of significant digits is too high for reporting aerosol pH, especially considering the standard deviations that are 20-50% of the actual value. Thus, for Line 243, I suggest reporting 4.2 +/- 2.2 and 5.7 +/- 1.4 for aerosol pH over the Gobi and Taklimakan Deserts, respectively. No need to use "~".*

**Response:** We thank the reviewer's suggestion. The number of significant digits is now reduced throughout the manuscript and "~" is deleted.

- *Lines 265-268, please explain why the doubling of NH3 emissions caused large or small changes in aerosol pH for different regions of China. This explanation would be aided by the maps of NVCs and NH3 emissions.*

**Response:** In fact, we have given the explanation in detail in the Discussion section in the original manuscript. We wrote mainly in the flow of presenting the results first and then discussing the reasons in the discussion part. As suggested, now we add Fig. S4 to show the spatial distributions of NVCs and NH₃ emissions to support our statements.

- ***Line 273, please include a figure showing that Cl- concentration is underestimated compared to observations.***

**Response:** Thanks for this suggestion. In the revised manuscript, we have plotted modeled and observed Cl⁻ concentration at Beijing site in Fig. S3c.

[Figure]

**Figure S3.** Comparison of simulated (a) $Ca^{2+}$ concentration (µg m⁻³), (b) $Na^+$ concentration (mEq m⁻³), and (c) Cl⁻ concentration (µg m⁻³) with observations (OBS; black line) for ORIG (blue line), CTL1 (green line), and CTL3 (red line) scenarios at Beijing site during the study period of 15 October 2014 - 02 November 2014, with the mean bias (MB), normalized mean bias (NMB) and average value (avg) given insert. MB and NMB are defined as $MB = \frac{1}{N}\sum_1^N C_m - C_o$

and $NMB = \frac{\sum_1^N C_m - C_o}{\sum_1^N C_o}$, where $C_m$ is the modeled value, $C_o$ is the observed value, and N is the number of paired model and observation data. $Mg^{2+}$ and $K^+$ are treated as charge-equivalent $Na^+$.

- ***Lines 273-275, deserves more description or explanation since the authors are advocating for future research on chloride emissions.***

**Response:** Thanks for your comment. Now we add more descriptions regarding the importance of chloride as "In addition, Cl⁻ is the precursor of reactive chloride species (e.g., Cl, ClNO₂, HOCl) that are important in atmospheric oxidation capacity (Wang et al., 2019; Wang et al., 2020b). For example, reactive chloride not only influences ozone and HOₓ concentrations, but also directly participates in atmospheric nitrate and sulfate production as oxidants (Wang et al., 2019; Wang et al., 2020). Recent studies (Gunthe et al., 2021; Chen et al., 2022) found that chloride is also important in

aerosol water uptake, playing an important role in the development of severe haze events."

- *Line 289. Is the analysis over Beijing averaged over several grid points or is it for a single location?*

**Response:** To compare simulated results with observations, the analysis over Beijing is from the individual grid box where the observational site is located.

- *Section 3.2, For the diurnal variation analysis, how did temperature and relative humidity vary? Is there a correlation between these state variables and aerosol pH? I would imagine that lower T, higher RH, which occur in the early morning, would have more aerosol water, diluting the hydrogen ion concentration.*

**Response:** For observations of aerosol compositions, hourly measurements are not available and in this study we are not focusing on the diurnal variation of aerosol pH. But to answer the reviewer's question, we plotted hourly data from the model results, i.e., the diurnal variations of temperature, relative humidity, aerosol water content, aerosol pH, and mole ratio of cation to anion (Fig. R3). In general, RH is higher during nighttime than daytime and reaches highest in the morning. T shows an inverse diurnal pattern with RH. AWC generally tracks the diurnal profile of RH, however, is probably also influenced by aerosol concentration, with its peak values at midnight. Aerosol pH has a similar pattern as AWC except for some special cases (e.g. a peak at 9:00), which could be explained by the mole ratio of cation to anion. These results indicate that in addition to meteorological conditions (thus AWC), the diurnal variation of aerosol pH may also driven by other factors including chemical compositions.

[Figure]

**Figure R3.** Diurnal profiles of modeled temperature [K], relative humidity [%], aerosol water content [μg m$^{-3}$], aerosol pH, and mole ratio of cation to anion at surface from CTL3meta scenario averaged for the study period of 15 October 2014 - 02 November 2014. Cation/anion are calculated as $\frac{Cation}{anion} = \frac{2 \times C_{SO4} + C_{NO3} + C_{Cl}}{C_{NH3} + C_{NH_4^+} + C_{Na+} + 2 \times C_{Ca^{2+}}}$, where C represents mole concentration.

- *Line 315, Since there were not NH3 observations made, what is the uncertainty in aerosol pH from ISORROPIA when using the empirical equation? This could be quantified by running ISORROPIA with +/-10% changes in NH3 concentration.*

**Response:** Upon the reviewers' suggestion, we test the effects of different NH$_3$ concentrations in aerosol pH predictions by varying NH$_3$ concentration by +/-10%. The results show that only +0.03 and -0.04 pH unit changes can be induced. Now we add these discussions as "In order to assess the effects of uncertainties in NH$_3$ concentration on aerosol pH

predictions, we also run ISORROPIA II with ±10% fluctuations in $NH_3$ concentration and find little changes (i.e., +0.03 and -0.04 pH unit) can be induced."

- *Lines 323-333. This paragraph reports results from Figure 4 but it does not provide any insight. Instead of giving details, I suggest discussing the meaning (or the point) of the results. For example, comparing CTL1 simulation to the default simulation could be written as, "When NVCs are increased, the aerosol pH increases by 0.9 on average with the largest increase occurring during clean periods because …. In contrast, when NH3 emissions are doubled, the aerosol pH increase was smaller (0.4 pH units) compared to CTL1 simulation because ….. With higher NH3 emissions, the pH increased more in the more polluted regions because …..". Please give explanations for each of the sensitivity cases.*

**Response:** Thanks for the comment. We agree with the reviewer that we should discuss what drives the different simulated aerosol pH under different scenario during haze events. To illustrate this, we revised this paragraph to include more discussions as follows:

"When NVCs are increased, the aerosol pH increases by 0.9 on average with the largest increase occurring during clean periods. This is likely because of the higher fraction of NVCs from primary aerosol in addition to insufficient neutralization by acid species due to their low concentrations from secondary formation compared to polluted periods. In contrast, when $NH_3$ emissions are doubled, the aerosol pH increase is smaller (0.4 pH units) compared to CTL1 simulation, which can be explained by the higher original pH and the semi-volatile nature of $NH_3$. With higher $NH_3$ emissions, the simulated pH increases more in more polluted periods. This is because aerosol pH is lower on more polluted conditions, which promotes more $NH_3$ shifting to aerosol phase to consume $H^+$, leading to increases in pH. Both increasing $Cl^-$ emission (CTL3 scenario) and changing phase state assumption (CTL3meta scenario) lead to negligible effects on pH in Beijing among all periods. For the two additional scenarios that incorporates heterogeneous S(IV) reactions, when considering ionic strength effects (CTL3het_Is scenario) little changes in the predicted $PM_{2.5}$ pH are seen, but more pronounced changes are seen when ionic strength effects are not taken into account (CTL3het_NoIs scenario). The latter case lead to the decreases in pH by 0.7 and 1.3 units for moderate and heavy pollution periods, respectively, due to the increased heterogeneous production of sulfate."

- *Lines 337-340. Why is aerosol pH sensitive to sulfate production in NCP? Why is the phase state assumption important to TD and GD regions? What are the "influencing factors" to the evolution of aerosol pH in a haze development cycle?*

**Response:** In NCP, high gas precursors (e.g., $SO_2$) from anthropogenic emissions together with high relative humidity promote more heterogeneous sulfate production, which cause $PM_{2.5}$ pH decrease significantly. In TD and GD regions where aerosols are in general solid due to low ambient RH, the metastable assumption is inappropriate and would lead to large decrease in pH as modeled. In fact, more detailed analysis about the reasons have been discussed in section 4.1.4 and 4.1.3, respectively. These sentences are just a brief summary of the main findings from Results.

We are sorry to make the last sentence misleading. Here "influencing factors" refers to emissions of nonvolatile cations (NVCs) and $NH_3$, aerosol phase state assumption, and heterogeneous production of sulfate. It is related to sensitivity of the $PM_{2.5}$ pH spatial variability, rather than in a haze cycle. Now we revise the statement to more clearly express our point as: "In the discussions as follows, we first analyze the sensitivity of $PM_{2.5}$ pH to factor such as NVCs emission, $NH_3$ emission, and etc., and then focus on the evolution of $PM_{2.5}$ pH in a haze development cycle in Beijing."

- *Line 345, please explain more how NVCs (or aerosol composition in general) affect aerosol water amount?*

**Response:** Aerosol composition exhibits different hygroscopicity, thus affecting aerosol water amount. We have revised the text to clarify this point: "NVCs are the alkaline components of aerosol which can neutralize sulfuric acid irreversibly and impact aerosol water amount through its effects on aerosol composition which regulates aerosol hygroscopicity, influencing aerosol pH both directly and indirectly."

- *Sections 4.1.1 and 4.1.2 would both benefit from showing maps of the NVCs and NH3 emissions. It would be interesting to see the juxtaposition of these cations.*

**Response:** Now these emission maps are added as Fig. S4 in the supporting material and are cited at appropriate positions to support our statements. And additional statement is added such as "However, in areas with high NVCs emissions (e.g. TD, GD; Fig. S4b), the increase in pH is not prominent (Fig. 2a) probably because in such regions the acidic species are already neutralized by NVCs which are alkaline."

- *Line 388, I like Figure 5. Panel a) clearly shows non-linearity within each region. However, is Figure 5b truly non-linear? Eyeballing each region seems to show a linear response but with different slopes between regions. Perhaps a fitting line(s) could be added to the figure.*

**Response:** This is a good question, but we afraid that the reviewer misunderstood what we meant. "nonlinearly" used here is to explain the relationship between changes in aerosol pH and $NH_3$ emissions. When $NH_3$ emissions are doubled, it has a notable impact on the pH of acidic aerosols but generates limited effects on alkaline aerosols. Regarding Fig. 5b, it shows a linear trend between pH changes and its original pH rather than $NH_3$ emission changes.

- *Lines 405-411, I suggest moving Figures S4 and S5 to the main text since they are discussed at length and provide important support to the conclusion that models need to represent both stable and metastable aerosols.*

**Response:** We agree with the reviewer that Figures S4 and S5 provide important support to our discussion. These two figures have been moved to main text as Figure 6 and Figure 7 in the revised manuscript.

- *Sections 4.1.3 and 4.1.4, It seems that producing Figure 5 plots for aerosol phase state and for sulfate production would be interesting. Could panels be added to Figure 5 for these additional simulations?*

**Response:** Thanks for your suggestion. Fig. 5 are revised now to include the CTL3meta (Fig. 5c) and CTL3het_noIs (Fig. 5d) results. The panels are cited in proper places in the text where they could support our previous statements and more texts are also added in the revised manuscript as "As shown in Fig. 5c, the small changes in water content could lead to a wide fluctuation in pH." and "Figure 5d shows that for these regions where $PM_{2.5}$ pH has an obvious response, the decrease of pH gets larger as original pH increases.".

[Figure]

**Figure 5.** Scatterplots of the surface PM$_{2.5}$ pH differences between (a) CTL1 and ORIG scenarios, (b) CTL2 and CTL1 scenarios, (c) CTL3meta and CTL3 scenarios, (d) CTL3het_NoIs and CTL3 scenarios vs. the corresponding original pH, separated by regions. Different scales are used.

- *Line 443, could the authors explain the interfacial chemistry a little bit more? What is it?*

**Response:** Here "interfacial chemistry" means chemical reactions occurring at the surface of deliquesced aerosol, which may obtain a larger reaction rate compared to bulk aqueous solutions. For example, Wang et al. (2021) found that Mn-catalyzed oxidation of SO$_2$ on the aerosol surface is orders of magnitude faster than in the bulk solution, mainly due to the different reaction mechanism, reaction space and surface. We have modified the text slightly to remind the contrast between interfacial chemistry and bulk chemistry as "Recent experimental studies suggest that interfacial chemistry at aerosol surfaces rather than in the bulk solutions may also be important for ambient sulfate formation, such as the newly proposed aerosol-phase acceleration for the Mn-catalyzed oxidation of S(IV) (Wang et al., 2021) and water-assisted interfacial reaction of NO$_2$ with SO$_3^{2-}$ (Liu and Abbatt, 2021)."

- *Lines 460-465, I think it is important to emphasize that aerosol pH during heavy pollution events remains < 5.0 noting that there is never a time when S(IV) + NO2 formation of S(VI) contributes significantly. Please add! Of course, this is for Beijing region and at dx=36km, which may average out some extreme situations where pH could go higher. When examining individual grid point temporal variations, are there times when pH > 5.0? What might happen if dx=12km (i.e. what might be the impact of the grid spacing on the results here)?*

**Response:** We agree with the reviewer's point on the role of NO$_2$ oxidation. We have revised the sentence to emphasize this as "These results suggest that PM$_{2.5}$ pH in Beijing under heavy haze conditions is likely moderate acidic (pH remains below 5.0), and thus the NO$_2$ oxidation pathway highly unlikely dominates in heterogeneous sulfate production."

Regarding the grid box, to compare simulated results with observations, Beijing data presented in the original manuscript are already from the individual grid point where the observation site is located. And for this individual grid point, aerosol pH is always much smaller than 5.0. We have also double checked all of the grid points in Beijing region, and only less than 5% of the time showing pH > 5.0 during heavy pollution days. In addition, we conducted sensitivity experiments at 12 km resolution and found that the impact of the grid spacing on the predicted pH is minor (please also see our response to your comment above). These results indicate that overall during these haze events, moderate acidic aerosol is dominant. Therefore, it is highly unlikely S(IV) + $NO_2$ formation of S(VI) contributes significantly to sulfate production in haze events.

- *Lines 470-480 or so. The discussion of Figure S8 is very useful. I suggest moving the figure into the main manuscript.*

**Response:** Considering the importance of Figure S8, we now move it to the main manuscript as Figure 9.

- *Lines 489-495, it may be beneficial to add a schematic describing what is said in the text here.*

**Response:** We agree that adding a schematic would be illustrative for the reader and therefore, we have added Fig. 11 to show the temporal evolution of $PM_{2.5}$ pH during haze cycle in Beijing.

[Figure]

**Figure 11.** The schematic plot of the temporal evolution of $PM_{2.5}$ pH during haze cycle in Beijing. The size of blue circles indicates the relative amount of aerosol water and the thickness of downward arrows indicates the relative strength of the process.

- *Line 499, why does the acid effect prevail over the dilution effect?*

**Response:** In the process of haze outbreak, AWC is greatly enhanced due to elevated ambient RH. The increased AWC can promote formation and ionization of acid species (such as sulfate and nitrate) and dilute the $H^+$ as well. The change of pH depends on the competition effect of both aspects. In view of the relationship that the modeled $PM_{2.5}$ pH decreases with increasing AWC, the acid effect might be greater than the dilution effect.

- *Section 4.3 provides useful information, but Pye et al. (2020) already presented these results. This section could be omitted or moved to the supplement.*

**Response:** We thank the reviewer for providing this suggestion. In response to this comment, Section 4.3 in the original manuscript is moved to Supplementary Information Text S1. The following sentences are added to Section 4.2 to provide a link.

"Despite their similar trend, overall ISORROPIA II predicts higher absolute pH values than that of MOSAIC with 1.1, 1.0 and 1.0 pH units higher during light, moderate and heavy pollution days, respectively, possibly due to the different thermodynamic representations such as activity coefficients and solution approach (see Text S1 for more details)."

- *Section 5. The Conclusions fall short in bringing the results to the greater context (or implications elsewhere). For example, instead of repeating "A priori assumption that aerosols are stable or metastable…", it could say that across China both stable and metastable state of aerosols exist, thus both states should be represented in regional and global models.*

**Response:** We thank reviewer this suggestion. In the end of the third paragraph of Conclusion, we further emphasize the insignificant role of S(IV) + NO$_2$ formation of S(VI) during haze events as "The moderately acidic aerosols under heavy haze conditions suggest that S(IV) oxidation by NO$_2$ is highly unlikely to contribute significantly to sulfate production in Beijing haze.". And following the suggestion by the reviewer, we revised the text about phase state assumption as "Across China both stable and metastable state of aerosols exist, thus both states should be represented in regional and global models.". Besides these, now we add more specific future needs in the last paragraph. Please see our response to the last comment below.

- *Section 5. The Conclusions did not say anything about the S(IV) + NO2 contribution to S(VI) yet this is part of the Abstract.*

**Response:** We thank reviewer for pointing this. Now we have added a sentence in Conclusion to emphasize this point as "The moderately acidic aerosols under heavy haze conditions suggest that S(IV) oxidation by NO$_2$ is highly unlikely to contribute significantly to sulfate production in Beijing haze."

- *Section 5. The Conclusions should also have more text about future needs. Why do we need more high-resolution observations? What is meant by "high resolution"? Do we need higher temporal resolution, or higher spatial resolution, or both? Would it be useful to have measurements throughout the boundary layer? What is the influence of boundary layer mixing on what is being measured? It seems that there is still a lot to be learned about winter-time haze and aerosol pH, and the authors have an opportunity to give their expert opinions on what should be prioritized.*

**Response:** We thank the reviewer for the comments, which are very useful for improving the manuscript. Now we add more statements about future needs as "More high temporal resolved observational datasets (e.g. hourly) are needed to help evaluate and understand the detailed evolution of pH during haze episodes as well as diurnal pattern of pH. Since observationally constrained pH is limited in terms of spatial coverage, more measurements need to be devoted to the regions where observations are rare or unavailable. In addition to aerosol composition, concurrent measurements of gas species subject to phase partitioning (e.g. HNO$_3$ and NH$_3$) will provide better constraints on acidity estimates. Measurements of size-resolved aerosol composition will also be useful to further evaluate MOSAIC predictions of aerosol pH from different size bins. What is more, future measurements can also consider to monitor throughout the boundary layer (e.g. from tall towers, mountain-based sites and aircraft) in order to provide insights into the vertical distribution of aerosol pH. The last, in-situ measurement technique of aerosol pH are desired to provide an improved understanding of aerosol pH and its effect on aerosol chemistry, and recently some approaches (e.g., Raman spectroscopy method (Cui et al., 2021; Li et al., 2022)) show the potential to do so in the future."

**Technical Comments**
- *Line 16, change "reported" to "report"*

**Response:** Corrected as suggested.

- *Line 18, add comma to read as "state assumption, and heterogeneous production"*

**Response:** Corrected as suggested.

- *Line 47, remove "for nowadays"*

**Response:** Thanks. We have removed it.

- *Lines 99-106, Lines 110-119, use present tense*

**Response:** We thank the referee for having drew our attention on this. We have corrected the sentences with present tense.

- *Line 110, cite Grell et al., 2005 and Fast et al., 2006 when introducing WRF-Chem model*

**Response:** Thanks for pointing it out. Now these papers are cited when WRF-Chem model is first mentioned (Line 59).

- *Section 2.3, please use present tense*

**Response:** The tenses have been corrected throughout this Section.

- *Line 187, just cite Fig. S3. No need to specify each of the panels.*

**Response:** Thanks for pointing this, and we have corrected it.

- *Line 206, remove "The last" and change "production as which" to "production for which"*

**Response:** The changes have been made accordingly.

- *Line 268, cite Figure S2.*

**Response:** The Figure (now Fig. S5) is cited now.

- *In the paper, Figure S3 is cited before Figure S2. Please review the text and order of figures.*

**Response:** Thanks for checking. The figures have been rearranged in order.

- *Line 278, "In particular, PM2.5 pH decreased by 1.9 for TD and 1.1 for GD, reducing aerosol pH values to 4.8 and 4.0, respectively, whereas the metastable state assumption had little impact …."*

**Response:** Thanks for your suggestion. Corrected.

- *Line 293, use present tense ("were" to "are")*

**Response:** Corrected as suggested.

- *Line 303, remove "On the other hand". I'm not sure what is being contrasted.*

**Response:** We have removed "On the other hand" to avoid confusion.

- *Line 315, remove "so"*

**Response:** Corrected as suggested.

- *Line 315, This sentence about NH3 observations should be in section 2.4. Please add the empirical equation to the supplemental text (or in the main text).*

**Response:** Thanks for your suggestion. We have moved this sentence to Section 2.4 and given the empirical equation. In addition, more discussion about the uncertainty in aerosol pH predictions made by the empirical equation is added as in the response to your comments above.

- *Section 3.2, Figure 4b is not discussed.*

**Response:** Thanks for noticing this. Now we add a brief discussion of Fig. 4b in Section 3.2 as "AWC generally tracks

the pattern of RH, with lowest water amount appearing during clean periods. Among all scenarios, ORIG predicts the lowest AWC. High abundance of AWC is seen in CTL3meta since metastable assumption normally predicts higher amount of water. The increased concentrations of sulfate in CTL3het_NoIs would enhance aerosol water uptake, resulting in more AWC. A detailed discussion of the correlation of AWC and pH during haze cycle can be found in Sect. 4.2."

- *Lines 337, reword the sentence removing "In addition". Maybe "For NCP where severe and frequent haze events occur, PM2.5 pH is very sensitive to the magnitude …."*

**Response:** We have reworded the text as suggested.

- *Lines 347-354 are stated in the Results section and do not need to be repeated here.*

**Response:** Thank you for the comment. We assume that the reviewer actually meant these results are mentioned in Section 2.3. The description in Section 2.3 is rather general and lacks of details. Therefore, these lines are merged into Section 2.3 to avoid repetition.

- *Line 362, change "minimum" to "minimal"*

**Response:** Corrected as suggested. Thank you.

- *Line 364, change "may" to "can". We know aerosol thermodynamics makes this sentence true. Same thing with Line 366 ("may" to "can"). You could also support these statements with supplement information using ISORROPIA calculations.*

**Response:** Thank you for raising this issue. We have made suggested changes.

- *Line 376, no need for both "In addition" and "also" in the same sentence. This occurs many times in the paper. Please proofread and try avoid using "in addition" and "also".*

**Response:** Thanks for checking. Now all of them are corrected in the revised manuscript.

- *Line 378, change "But" to "However,"*

**Response:** Now this sentence is removed for reducing repeated contents.

- *Line 416, change to "all size bins"*

**Response:** Corrected as suggested.

- *Line 421, change "might be" to "is" as you just spent two paragraphs showing the consequence of this assumption.*

**Response:** We agree and revised it accordingly.

- *Line 431, could you be more explicit about "differ by regions"? Where would the sulfate production be buffered?*

**Response:** Thanks for your comment. Now we give an example to clarify it as "For example, relatively prominent sulfate production occurs in the south part of Jiangxi Province, whereas the corresponding decrease in pH is less obvious, which may be partially offset by the buffering effect of excess ammonia."

- *Line 438, the authors may want to note that H2O2 oxidation may be small because of low OH during the time of year investigated for the study.*

**Response:** Thanks for this comment. Here we just note the $H_2O_2$ concentration is low during the study period, but we have not explored the reason why $H_2O_2$ is low and if it is indeed related to the low OH. The atmospheric oxidants (e.g. OH) are supposed to be in low concentrations during wintertime because of less active photochemistry. However, some

field studies indicate, the reactivity of OH is high during winter haze events in Beijing, even similar to the results observed in summer, which suggests that strong gas-phase oxidation still occurs in wintertime (Lu et al., 2018; Ma et al., 2019; Slater et al., 2020; Tan et al., 2018).

- ***Line 450, please state why CTL3meta was chosen for further analysis instead of any of the other simulations.***

**Response:** Thanks for pointing it out. Now we add the clarification of the selection of CTL3meta in the Section 4.2 as "CTL3meta scenario is selected because this scenario shows a better agreement with observations on PM$_{2.5}$ compositions and allows us to make a fair comparison with ISORROPIA II in which the metastable state is also assumed."

- ***Line 460, change to "under heavy pollution events …. was 3.6 +/- 0.5".***

**Response:** Corrected as suggested.

- ***Line 485, delete "also"***

**Response:** Deleted.

- ***Line 486, change to "high on more polluted"***

**Response:** Corrected as suggested.

- ***Line 512, change to "both models using the same"***

**Response:** Corrected as suggested.

- ***Table 1, add vertical domain and grid (i.e., number of vertical levels, how many levels in the boundary layer). Also add information about observational nudging of the meteorology.***

**Response:** Thanks for your suggestion. Table 1 has been updated to include all these information, and a typo in Line 173 "139 (west-east) × 148 (south-north)" was corrected to "138 (west-east) × 149 (south-north)".

Table1 is as following:

**Table 1.** Summary of model configurations.

| Description | | Selection |
|---|---|---|
| Horizontal grid spacing | | 36 km |
| Vertical levels | | 41 (roughly 8 layers below 1 km) |
| Grid dimensions | | 149 × 138 |
| Aerosol scheme | | MOSAIC 8 bin |
| Gas-phase chemistry | | CBM-Z |
| Long wave Radiation | | RRTMG |
| Short wave Radiation | | RRTMG |
| Cloud Microphysics | | Morrison 2-moment |
| Cumulus Cloud | | Grell-Devenyi |
| Planetary boundary layer | | YSU |
| Land surface | | Noah land-surface model |
| Grid nudging | Nudging variables | u and v component wind, air temperature, water vapor mixing ratio |
| | Applied layers | Layers above the PBL |
| | Nudging timescale | 6 h |

- *Figure 1a. The six sub-regions are hard to read. Try using gray for the geography and a darker color for the acronyms.*

**Response:** Figure 1a is revised following your suggestion.

- *Figure 2, change CLT to CTL (check all figures for this misspelling)*

**Response:** Thanks for your careful check. We carefully went through all figures to ensure there are no misspellings.

- *Figure 3 panels are in a different order than Figure 2*

**Response:** Thank the reviewer for the detailed check. Figure 3 panel order is now revised to keep consistent order with other figures.

- *Figure 4b, could WRF RH line be black (to be consistent with observations) and maybe dashed.*

**Response:** WRF RH data are now shown by black dashed line.

- *Figure 7, are these results for all the simulated days or for just the polluted days?*

**Response:** As stated in the caption of Fig. 7 (now Fig. 10) that "during the study period of 15 October 2014 - 02 November 2014", these results are for all the simulated days.

- *Table S2, change to "ranges and mean" because of the order of the columns.*

**Response:** Corrected as suggested. Thanks.

- *Table S3, where are the "decreasing regions"? Be more descriptive. Also add the time period averaged over for these results.*

**Response:** Thanks for your careful check. Now the "pH- decreasing regions" is marked by blue box in Fig. 7. In the main text we have added the following sentence: "We select one area (denoted by the blue box in Fig. 7) in the pH-decreasing regions to discuss the characteristics in detail."

And the caption to Table S3 has been clarified as below:

"The concentrations (in unit of $\mu mol \ m^{-3}$) of major $PM_{2.5}$ components for each bin (01-06) averaged over the pH-decreasing regions (denoted by the blue box in Fig. 7) in CTL3meta scenario during the study period of 15 October 2014 to 02 November 2014."

- *Figure S3a. Should the observations be plotted as markers for when the measurements are taken (like in S3b)? A line for the observations could be misleading.*

**Response:** Thanks for your suggestion. Now observations in Figure S3a are plotted as markers like in S3b.

- *Figure S3. The last two sentences are not clear. Why are they included in the figure caption?*

**Response:** To avoid confusion, we decided to remove the last two sentences from figure caption.

- *Figure S4. Is it surface relative humidity or RH at 2m?*

**Response:** Thanks for pointing this out. It is RH at 2m. Now we clarify it in caption of Fig. S4 (now Fig. 6).

**Reference**

Chen, Y., Wang, Y., Nenes, A., Wild, O., Song, S., Hu, D., Liu, D., He, J., Hildebrandt Ruiz, L., Apte, J. S., Gunthe, S. S., and Liu, P.: Ammonium Chloride Associated Aerosol Liquid Water Enhances Haze in Delhi, India, Environ. Sci. Technol., https://doi.org/10.1021/acs.est.2c00650, 2022.

Cui, X., Tang, M., Wang, M., and Zhu, T.: Water as a probe for pH measurement in individual particles using micro-Raman spectroscopy, Anal. Chim. Acta, 1186, 339089, https://doi.org/https://doi.org/10.1016/j.aca.2021.339089, 2021.

Gunthe, S. S., Liu, P., Panda, U., Raj, S. S., Sharma, A., Darbyshire, E., Reyes-Villegas, E., Allan, J., Chen, Y., Wang, X., Song, S., Pöhlker, M. L., Shi, L., Wang, Y., Kommula, S. M., Liu, T., Ravikrishna, R., McFiggans, G., Mickley, L. J., Martin, S. T., Pöschl, U., Andreae, M. O., and Coe, H.: Enhanced aerosol particle growth sustained by high continental chlorine emission in India, Nat. Geosci., 14, 77-84, https://doi.org/10.1038/s41561-020-00677-x, 2021.

He, P., Alexander, B., Geng, L., Chi, X., Fan, S., Zhan, H., Kang, H., Zheng, G., Cheng, Y., Su, H., Liu, C., and Xie, Z.: Isotopic constraints on heterogeneous sulfate production in Beijing haze, Atmos. Chem. Phys., 18, 5515-5528, https://doi.org/10.5194/acp-18-5515-2018, 2018.

Li, L.-F., Chen, Z., Liu, P., and Zhang, Y.-H.: Direct Measurement of pH Evolution in Aerosol Microdroplets Undergoing Ammonium Depletion: A Surface-Enhanced Raman Spectroscopy Approach, Environ. Sci. Technol., 56, 6274-6281, https://doi.org/10.1021/acs.est.1c08626, 2022.

Lu, K., Guo, S., Tan, Z., Wang, H., Shang, D., Liu, Y., Li, X., Wu, Z., Hu, M., and Zhang, Y.: Exploring atmospheric free-radical chemistry in China: the self-cleansing capacity and the formation of secondary air pollution, National Science Review, 6, 579-594, https://doi.org/10.1093/nsr/nwy073 2018.

Ma, X., Tan, Z., Lu, K., Yang, X., Liu, Y., Li, S., Li, X., Chen, S., Novelli, A., Cho, C., Zeng, L., Wahner, A., and Zhang, Y.: Winter photochemistry in Beijing: Observation and model simulation of OH and HO2 radicals at an urban site, Sci. Total Environ., 685, 85-95, https://doi.org/https://doi.org/10.1016/j.scitotenv.2019.05.329, 2019.

Seaman, N. L., Stauffer, D. R., and Lario-Gibbs, A. M.: A Multiscale Four-Dimensional Data Assimilation System Applied in the San Joaquin Valley during SARMAP. Part I: Modeling Design and Basic Performance Characteristics J. Appl. Meteorol., 34, 1739-1761, https://doi.org/10.1175/1520-0450(1995)034<1739:Amfdda>2.0.Co;2, 1995.

Slater, E. J., Whalley, L. K., Woodward-Massey, R., Ye, C., Lee, J. D., Squires, F., Hopkins, J. R., Dunmore, R. E., Shaw, M., Hamilton, J. F., Lewis, A. C., Crilley, L. R., Kramer, L., Bloss, W., Vu, T., Sun, Y., Xu, W., Yue, S., Ren, L., Acton, W. J. F., Hewitt, C. N., Wang, X., Fu, P., and Heard, D. E.: Elevated levels of OH observed in haze events during wintertime in central Beijing, https://doi.org/10.5194/acp-2020-362, 2020.

Stauffer, D. R., and Seaman, N. L.: Use of Four-Dimensional Data Assimilation in a Limited-Area Mesoscale Model. Part I: Experiments with Synoptic-Scale Data, Monthly Weather Review, 118, 1250-1277, https://doi.org/10.1175/1520-0493(1990)118<1250:Uofdda>2.0.Co;2, 1990.

Tan, Z., Rohrer, F., Lu, K., Ma, X., Bohn, B., Broch, S., Dong, H., Fuchs, H., Gkatzelis, G. I., Hofzumahaus, A., Holland, F., Li, X., Liu, Y., Liu, Y., Novelli, A., Shao, M., Wang, H., Wu, Y., Zeng, L., Hu, M., Kiendler-Scharr, A., Wahner, A., and Zhang, Y.: Wintertime photochemistry in Beijing: observations of ROx radical concentrations in the North China Plain during the BEST-ONE campaign, Atmos. Chem. Phys., 18, 12391-12411, https://doi.org/10.5194/acp-18-12391-2018, 2018.

Wang, W., Liu, M., Wang, T., Song, Y., Zhou, L., Cao, J., Hu, J., Tang, G., Chen, Z., Li, Z., Xu, Z., Peng, C., Lian, C., Chen, Y., Pan, Y., Zhang, Y., Sun, Y., Li, W., Zhu, T., Tian, H., and Ge, M.: Sulfate formation is dominated by manganese-catalyzed oxidation of SO2 on aerosol surfaces during haze events, Nat. Commun., 12, 1993, https://doi.org/10.1038/s41467-021-22091-6, 2021.

Wang, X., Jacob, D. J., Eastham, S. D., Sulprizio, M. P., Zhu, L., Chen, Q., Alexander, B., Sherwen, T., Evans, M. J., Lee, B. H., Haskins, J. D., Lopez-Hilfiker, F. D., Thornton, J. A., Huey, G. L., and Liao, H.: The role of chlorine in global tropospheric chemistry, Atmos. Chem. Phys., 19, 3981-4003, https://doi.org/10.5194/acp-19-3981-2019, 2019.

Wang, X., Jacob, D. J., Fu, X., Wang, T., Breton, M. L., Hallquist, M., Liu, Z., McDuffie, E. E., and Liao, H.: Effects of Anthropogenic Chlorine on PM2.5 and Ozone Air Quality in China, Environ. Sci. Technol., 54, 9908-9916,

https://doi.org/10.1021/acs.est.0c02296, 2020.

Zhao, C., Liu, X., Leung, L. R., Johnson, B., McFarlane, S. A., Gustafson Jr, W. I., Fast, J. D., and Easter, R.: The spatial distribution of mineral dust and its shortwave radiative forcing over North Africa: modeling sensitivities to dust emissions and aerosol size treatments, Atmos. Chem. Phys., 10, 8821-8838, https://doi.org/10.5194/acp-10-8821-2010, 2010.

---

## Author Response (AR2)

**Referee #2**

*I am really impressed by how responsive the authors were to my and Reviewer 1's comments. This is now an excellent paper and should be accepted for publication. I suggest a couple of minor changes that I hope could be implemented in the copy editing phase.*

**Response:** We would like to thank the reviewer for the time reviewing this manuscript and providing supportive comments. Please find our point-by-point responses below.

**Specific Comments**

- *I recommend adding a comment that dx=12km simulations were conducted and results did not differ from those conducted at dx=36km.*

**Response:** Now we add a comment in the end of the third paragraph of Conclusion as "Sensitivity experiments were also conducted at finer resolution (12km) and the results did not differ from those conducted at 36km resolution."

**Technical Comments**

- *I suggest citing the schematic (Figure 11) on line 545 of the track-changes version, where it says, "can be explained as follows" to "can be explained as follows and shown in Fig. 11"*

**Response:** Thanks for your suggestion. Corrected.

- *Table 1. Include the reanalysis dataset used for nudging.*

**Response:** Table 1 has been updated as suggested.